# Extensive Cryptic Diversity and Ecological Associations Uncovered among Mexican and Global Collections of *Naegleria* and *Vermamoeba* Species by 18S Ribosomal DNA, Internal Transcribed Spacer, and Cytochrome Oxidase Subunit I Sequence Analysis

Juan M. Zurita-Artaloitia,[a] Javier Rivera,[a] (ID) Pablo Vinuesa[a]

[a]Centro de Ciencias Genómicas, Universidad Nacional Autónoma de México, Cuernavaca, Morelos, Mexico

Juan M. Zurita-Artaloitia and Javier Rivera contributed equally to this article. Author order was determined in order of decreasing seniority.

**ABSTRACT** Free-living amoebae (FLA) are phagocytic protists that play crucial roles in microbial communities as significant microbial grazers. However, our current knowledge of their diversity, ecology, and population genetic structures is marginal due to the shallow and biased sampling of ecosystems and the use of few, poorly resolving molecular markers. Thirty-two FLA were isolated from soil and water samples collected across representative ecosystems of the State of Morelos in Central Mexico, including the drinking water distribution system (DWDS) from the state capital. We classified our isolates as members of *Acanthamoeba*, *Vermamoeba*, *Naegleria*, and *Tetramitus* by 18S ribosomal DNA (rDNA) sequencing. *Vermamoeba* isolates were recovered exclusively from the DWDS samples. In contrast, *Naegleria* strains displayed a broad distribution in soil and water samples across the natural ecosystems. We used a combination of phylogenetic and population genetic analyses of internal transcribed spacer (ITS) and cytochrome oxidase subunit I (COI) sequences from our isolates and a comprehensive set of reference sequences to analyze the currently known diversity of *Naegleria* spp. Significant associations were uncovered between the most prevalent lineages of *Naegleria* and *Vermamoeba* and broad ecological and geographical variables at regional and global levels. The population structure and cryptic diversity within the *Naegleria galeacystis-Naegleria americana* and *Vermamoeba vermiformis* species complexes were thoroughly analyzed. Our results prove that the genus *Vermamoeba*, which was previously thought to consist of only one species, actually encompasses at least seven widely distributed species, as indicated by consistent evidence from Bayesian phylogenetics, two species-delimitation programs, and population genetics analyses.

**IMPORTANCE** Our study sheds new light on the population genetic structure of *V. vermiformis* and diverse *Naegleria* species. Using improved molecular markers and advanced analytical approaches, we discovered that *N. americana*, previously considered a single species, actually contains multiple distinct lineages, as revealed by COI sequencing. These lineages are highly differentiated, with little gene flow between them. Our findings demonstrate that the genus *Vermamoeba* holds multiple cryptic species, requiring a significant taxonomic revision in light of multilocus sequence analyses. These results advance our understanding of the ecology, molecular systematics, and biogeography of these genera and species complexes at both regional and global scales. This study has significant implications for diagnosing amoebal infections and evaluating health risks associated with FLA in domestic and recreational waters.

**KEYWORDS** free-living amoebae, species delimitation, barcoding primers, molecular systematics, Mexico, biogeography, ecology, drinking water distribution system, ASAP, mPTP, Bayesian phylogenetics, population genetics, COI, ITS

Address correspondence to Pablo Vinuesa, vinuesa@ccg.unam.mx.

The authors declare no conflict of interest.

10.1128/spectrum.03795-22 **1**

Free-living amoebae (FLA) comprise a diverse and ancient assemblage of testate and naked predatory protists that feed on other microorganisms by phagocytosis, playing an essential role in shaping microbial communities (1, 2). However, some microbes, including diverse opportunistic pathogens, have developed various molecular strategies to evade amoebal digestion, eventually using them as replication or survival niches (3–5) Over evolutionary time, these interactions developed into a notorious diversity of symbioses between protists, bacteria, and archaea (6). Amoeba move by extending and retracting cellular projections known as pseudopods, which define their characteristic ameboid movement. Based on these attributes, ameboid protists were initially placed in the Sarcodina group (7). However, phylogenetic analysis of 18S rRNA gene sequences (8, 9) and more recent large-scale multilocus phylogenomic studies revealed the polyphyly and deep phylogenetic distances that separate the currently known FLA lineages (10–12). As a result, Sarcodina was split into multiple supergroups (8, 13), highlighting the magnitude of the challenge faced by microbial ecologists and systematists interested in performing comprehensive diversity studies with FLA (3, 14, 15).

Species of FLA are generally thought to be broadly distributed (16, 17). However, poorly resolving molecular markers, combined with a shallow sampling of a few ecosystems, constrain our knowledge of the diversity and ecological features of FLA on local, regional, and global geographical scales (18–20). The most frequently used markers to study the molecular diversity of environmental FLA target internal segments of the small subunit (SSU) rRNA gene or complete ribosomal internal transcribed spacer (ITS) regions. Despite the popularity of cytochrome oxidase subunit I (COI or *cox1*) barcoding for animal and fungal diversity studies (21), it has only been marginally used to analyze the diversity of protists and FLA (14).

In this work, we performed a multilocus sequence analysis of the species diversity of FLA across representative ecosystems of the State of Morelos (Central Mexico), including conserved, perturbed, and urban sites. We used evidence from nuclear (18S ribosomal DNA [rDNA] and ITS1-5.8S-ITS2 rDNA region) and mitochondrial COI loci, coupled with complementary phylogenetic and population genetic methods to delimit species, revise the taxonomy of *Naegleria* and *Vermamoeba*, test ecological hypotheses, and refine our knowledge on the global distribution patterns of selected taxa. We developed primers to generate standardized barcoding markers to achieve these goals. Given the previously mentioned challenges in studying the diversity of environmental FLA, the two-step barcoding strategy suggested by the Consortium for the Barcode of Life (CBOL) Protist Working Group (14) was implemented. We first evaluated primers F-566/R-1200, which amplify eukaryotic SSU rDNA fragments of about 600 to 650 bp, spanning the V4 and V5 regions (22). To the best of our knowledge, they have not been used previously to study the diversity of environmental isolates of FLA. These primers generated high-quality amplicons and sequences for all our isolates. We carefully selected closely related reference sequences from the public databases for phylogenetic analysis, which allowed the unambiguous identification of all our isolates at the genus level, including the species in most instances. The amplified region was informative enough to uncover a highly significant association between the recovered taxa of FLA and ecological variables. In a second step, we adapted standard and widely used primers targeting the ITS1-5S-ITS2 rDNA region (JITS) (23) and COI genes (LCO1490/HC02198) (24) to improve their binding on heterolobosean Tetramitia and amoebozoan Discosea and Tubulinea, respectively, which were the higher-level taxa (13) found among our isolates. Combined phylogenetic and population genetic analyses of the new sequences generated in this study and selected global reference sequences confirmed the identification of the isolates at the species level based on the current taxonomy. However, these markers revealed extensive intraspecific variation within *Naegleria americana-Naegleria galeacystis* and *Vermamoeba vermiformis*. Bayesian phylogenetics and single-locus species delimitation methods were combined with population genetic structure analyses to evaluate alternative hypotheses of species borders, revealing for the first time the existence of multiple species-like units within both lineages. We also provide a comprehensive analysis of the

**TABLE 1** Location and ecological description of the sampling sites

| Site no. | Ecosystem/bioclimatic zone | Coordinates | Altitude (m.a.s.l.) | Habitat and site description[a] | Sample identifier |
|---|---|---|---|---|---|
| 1 | Seasonally dry tropical forest | N 18°51′14.6″, W 99°13′19.8″ | 1,240 | Water column and sediment of the Apatlaco River (polluted) at Temixco, along perturbed riparian gallery forest with *Salix bonplandiana* (ahuejote), *Salix humboldtii* and *Taxodium huegellii* (ahuehuete), *Ficus cotinifolia* (amate negro), and *Inga vera* (aguatope). | TS*, TA* |
| 2 | Pine-oak forest | N 18°59′06″, W 99°14′05″ | 1,917 | Humid upper soil horizon, rich in organic matter, just 5–10 cm below the superficial mold of pine needles and oak leaves. | *MJS* |
| 3 | Pine-fir forest | N 18°58′32″, W 99°20′31″ | 2,327 | Humid upper soil horizon, rich in organic matter, just 5–10 cm below the superficial mold. | S* |
| 4 | Seasonally dry tropical forest | N 18°48′17.8″, W 99°15′53.3″ | 1,112 | Upper soil horizon. | XOCH* |
| 5 | Seasonally dry tropical forest | N 18°43′47″, W 99°06′46″ | 954 | Water column and aquatic plant *Ludwigia palustris* leaves growing in Las Estacas River (Natural Park). | EAP1L* |
| 6 | Municipal water distribution system | N 18°57′41″, W 99°14′46″ | 1,750 | Different filters of a domestic water purifying system. | FER*, FCA*, SFI* |
| 7 | Municipal water distribution system | N 18°58′51.0″, W 99°14′41″ | 1,829 | Domestic tap water (shower), site A. | JR* |
| 8 | Municipal water distribution system | N 18°58′22″, W 99°14′46″ | 1,864 | Domestic tap water (shower), site B. | DR* |

[a]Ecological description of ecosystems partly based on reference 103.

known diversity of *Naegleria* ITS sequences and used them to identify significant associations between certain species complexes, their geographic distribution, and other ecological variables. The molecular markers, state-of-the-art analytical approaches, and comprehensive analyses presented here significantly contribute to advancing our understanding of the biogeography, molecular systematics, and ecology of these broadly distributed but poorly characterized species complexes. We discuss the implications of our findings for the taxonomic, ecological, and clinical fields.

## RESULTS

**A collection of free-living amoebae recovered from water and soil samples collected across ecosystems of Morelos, Central Mexico.** We sampled soil and aquatic habitats from eight sites (Table 1) located along an altitudinal and bioclimatic gradient within the three major ecosystems found in the State of Morelos, Central Mexico. We enriched to purity 32 isolates of FLA. Eighteen isolates (56%) were recovered from different sites of the drinking water distribution system (DWDS) of Cuernavaca, the state capital. The remaining 14 isolates were recovered from natural sites (Table 2).

**Maximum likelihood (ML) phylogeny of SSU rDNA sequences from F-566/R-1200 amplicons allows unambiguous identification at the genus level of the 32 Mexican isolates recovered in this study.** Figure 1 shows the best ML tree for SSU rDNA sequences found by an IQTree2 search. The two main clades resolved (I and II) correspond to the deeply divergent Amoebozoa and Discoba supergroups (13). Higher taxonomic ranks in this work follow those proposed by Adl and colleagues (13). Using carefully selected close reference sequences collected from NCBI, we could unambiguously classify all our isolates at the genus level. Species identification was possible in most instances. Clade I (Amoebozoa) splits into two subclades corresponding to the phyla Discosea (25) and Tubulinea (26), respectively. Discosea holds reference *Acanthamoeba* sequences along with two Mexican isolates closely related (*p*-distances of 0.0012 and 0.0073 substitutions/site or 0.12% and 0.73% sequence divergence) to a sequence from *Acanthamoeba* genotype T13, collected in the Tibetan plateau (27). Given this low divergence and the fact that the F-566/R-1200 amplicons overlap entirely with the ASA.S1

**TABLE 2** List of the 32 Mexican isolates obtained and analyzed in this study, along with their classification, sample metadata

| Species[a] | Isolate | Sample origin[b,c] | Sampling site no. | Collection date[d] |
|---|---|---|---|---|
| *Acanthamoeba* genotype T13 | 4MJS2 | Pine-oak forest soil, CCG, Chamilpa | 2 | 10/19 |
| *Acanthamoeba* genotype T13 | 5HKMJS2 | Pine-oak forest soil, CCG, Chamilpa | 2 | 10/19 |
| *Vermamoeba vermiformis* complex | FER1 | Domestic water filtration unit | 6 | 02/20 |
| *Vermamoeba vermiformis* complex | FCA1 | Domestic water filtration unit | 6 | 02/20 |
| *Vermamoeba vermiformis* complex | FCA2 | Domestic water filtration unit | 6 | 02/20 |
| *Vermamoeba vermiformis* complex | FCA3 | Domestic water filtration unit | 6 | 02/20 |
| *Vermamoeba vermiformis* complex | FCA4 | Domestic water filtration unit | 6 | 02/20 |
| *Vermamoeba vermiformis* complex | SF1-1 | Domestic water filtration unit | 6 | 02/20 |
| *Vermamoeba vermiformis* complex | SF1-2 | Domestic water filtration unit | 6 | 02/20 |
| *Vermamoeba vermiformis* complex | SF1-3 | Domestic water filtration unit | 6 | 02/20 |
| *Vermamoeba vermiformis* complex | SF1-4 | Domestic water filtration unit | 6 | 02/20 |
| *Vermamoeba vermiformis* complex | DR1 | Showerhead biofilm | 8 | 09/19 |
| *Vermamoeba vermiformis* complex | DR2 | Showerhead biofilm | 8 | 09/19 |
| *Vermamoeba vermiformis* complex | DR3 | Showerhead biofilm | 8 | 09/19 |
| *Vermamoeba vermiformis* complex | DR4 | Showerhead biofilm | 8 | 09/19 |
| *Vermamoeba vermiformis* complex | DR6 | Showerhead biofilm | 8 | 09/19 |
| *Vermamoeba vermiformis* complex | DR7 | Showerhead biofilm | 8 | 09/19 |
| *Vermamoeba vermiformis* complex | DR8 | Showerhead biofilm | 8 | 09/19 |
| *Vermamoeba vermiformis* complex | DR9 | Showerhead biofilm | 8 | 09/19 |
| *Vermamoeba vermiformis* complex | JR1 | Showerhead biofilm | 7 | 09/19 |
| *Naegleria americana* complex | S5 | Pine-fir forest soil, Las Cascadas | 3 | 09/20 |
| *Naegleria americana* complex | S6 | Pine-fir forest soil, Las Cascadas | 3 | 09/20 |
| *Naegleria americana* complex | XOCH1-2 | SDT Forest soil, Xochitepec | 4 | 09/20 |
| *Naegleria americana* complex | XOCH2-3 | SDT Forest soil, Xochitepec | 4 | 09/20 |
| *Naegleria americana* complex | XOCH1-3 | SDT Forest soil, Xochitepec | 4 | 09/20 |
| *Naegleria americana* complex | TS3-1 | Sediment, Apatlaco river | 1 | 03/20 |
| *Naegleria americana* complex | TS3-2 | Sediment, Apatlaco river | 1 | 03/20 |
| *Naegleria americana* complex | EAP1L | *Ludwigia palustris* leaves, Las Estacas | 5 | 03/21 |
| *Naegleria gruberi* | 3HKMJS1 | Pine-oak forest soil, CCG | 2 | 10/19 |
| *Naegleria* sp. (*N. clarki* complex) | TA37-1 | Water column, Apatlaco river | 1 | 03/20 |
| *Tetramitus* sp. (new lineage) | TS37-1 | Sediment, Apatlaco river | 1 | 03/20 |
| *Tetramitus* sp. (new lineage) | TS37-2 | Sediment, Apatlaco river | 1 | 03/20 |

[a]Taxonomic binomial correspond to currently accepted species. However, multiple evolutionary significant units were uncovered within the *N. americana/N. galeacystis* and *V. vermiformis* species complexes, as detailed in the main text.
[b]SDT, Seasonally dry tropical forest.
[c]See Table 1 for a detailed ecological description of the sampling sites.
[d]Mo/yr.

region used for *Acanthamoeba* genotyping (28), our *Acanthamoeba* isolates were classified as members of genotype T13 (29), frequently found in soils (30). Both isolates originated from mixed pine-oak forest soil samples collected at site 2 (Table 1). The remaining 19 isolates in clade I grouped along with representatives of the class Echinamoebida (25). They were all collected from the Cuernavaca, Mexico, drinking water distribution system (DWDS) (Table 1) and clustered tightly (0.0% to 0.5% divergence) and with high support with *V. vermiformis* (formerly *Hartmannella vermiformis*) reference sequences. *Vermamoeba* is currently considered a monospecific genus (17, 31).

The isolates grouped in clade II (supergroup Discoba) are members of the family Vahlkampfiidae. Two of the Mexican isolates in this clade belong to the genus *Tetramitus*, clustering closely (0.28% divergence) with the sequence from the type strain of *Tetramitus dokdoensis* (32), an amoeboflagellate isolated from a freshwater pond on Dokdo Island, South Korea. The remaining isolates are members of the genus *Naegleria*, which form a tight, perfectly supported clade with multiple reference sequences. Isolate 3HKMJS1 shares the same partial 18S rDNA sequence with *Naegleria gruberi* strain EGB (33), while strain TA37-1 clusters closely with *Naegleria clarki* sequences. The remaining eight isolates are most similar (0% to 0.63% substitutions) to the reference sequence for *N. americana* strain NG 006 (34). Most *Naegleria* isolates originated from soil or water samples from the warmer bioclimatic zone occupied by seasonally dry tropical forests. However, the *N. gruberi*-like isolate 3HKMJS1 was recovered from the pine-oak soil at site 2 (Table 1), and isolates S5

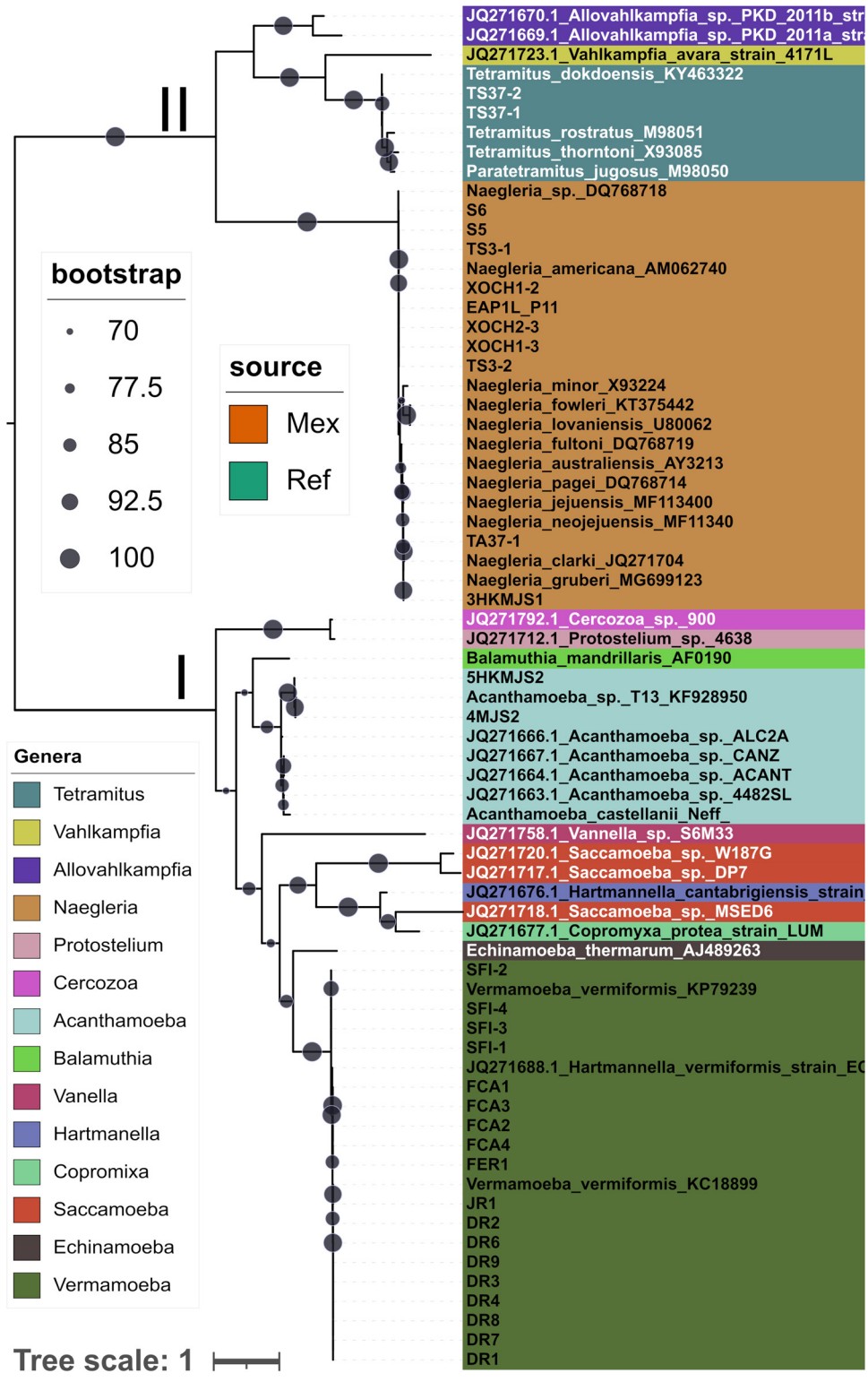

**FIG 1** The maximum likelihood tree (ln*L* = −8,955.425) inferred from 18S rRNA sequences shows the phylogenetic placement of the 32 Mexican isolates. The tree was estimated under the best-fitting TIM2+F+R3 model, selected using the Bayesian information criterion (BIC) out of 484 competing DNA models. The sequences were aligned with SINA, with borders manually trimmed. The resulting matrix had 70 sequences with 721 columns, 661 distinct patterns, 560 parsimony-informative, 134 singleton, and 26 constant sites. Five sequences failed the composition chi-square test ($P < 5\%$; df = 3). Total tree length, 12.464. Numbers on internal nodes correspond to UFBoot support values. The bar indicates the number of expected substitutions per site under the TIM2+F+R3 model. The color strip indicates the source of the sequences (Ref, GenBank; Mex, Mexican isolates). A compressed tar file with the tree and alignment is available online at https://doi.org/10.6084/m9.figshare.22011095.

and S6 originated from pine-fur forest soil (upper organic horizon) at site 3, both at higher elevations, with a temperate climate (Table 1).

In summary, isolates from the genera *Naegleria* and *Vermamoeba* were the most prevalent in our collection. *Naegleria* isolates were recovered from soil and water samples collected along the altitudinal gradient and spanning diverse natural ecosystems. In contrast, *Vermamoeba* isolates were isolated exclusively from the DWDS samples.

**A highly significant association between genetic and ecological variables for the novel Mexican *Naegleria* and *Vermamoeba* strains.** We challenged the (null) hypothesis that genera of FLA are homogeneously distributed across ecosystems and habitats using a three-way association test between the genus and the ecological variables "ecosystem" and "habitat," as defined in Table 1. We included only the two genera with ≥10 isolates. The highly significant test strongly rejected the null ($P < 2.22e$-16). Figure 2A reveals the tight associations of *V. vermiformis* with samples from the Cuernavaca DWDS and those of *Naegleria* isolates with river water columns, sediments, and soils collected along the natural ecosystems (Table 1). Permutation tests with 10e + 5 replicates for conditional independence between the genus+ecosystem and genus+habitat variable pairs were very highly significant [$f(x) = 2.9519$, $P = 9e$-05; $f(x) = 2.7329$, $P = 0.00051$; respectively], strongly rejecting the null of independence. This result indicates that the sampled organisms from both genera have strongly differentiated environmental distributions, which may result from distinct habitat preferences or adaptations.

**Ecological associations and global diversity of *Naegleria* species assessed by phylogenetic analysis of a comprehensive set of ITS1-5.8S-ITS2 sequences from GenBank.** ITS1-5.8S-ITS2 rDNA sequences are currently the favored markers for vahlkampfid species identification (23), including the description of new ones, defined as strains sharing a distinct ITS sequence (34). Figure 3 shows the ML phylogeny of the currently known diversity of *Naegleria* species, based on 442 selected ITS sequences found in GenBank (as of August 2022) and 10 new ones generated in this study for the Mexican *Naegleria* isolates (Table 2). Theis data set contains 197 (43.58%) unclassified (*Naegleria* sp.) and 255 (56.42%) classified sequences, including 13 type strains (annotated in Fig. 3) and 9 of our Mexican isolates. We identified 40 lineages (Fig. 2B) and defined 28 phylogenetic clusters as highly supported (UFBootstrap, >90%) terminal clades with a long subtending branch. They hold multiple sequences for a single named or very closely related taxonomic species ($n = 23$) or tight terminal clusters of unnamed sequences ($n = 5$; new1 . . . new5), as shown in Fig. 2B and defined in Table S1 posted online at https://doi.org/10.6084/m9.figshare.21200467. In addition, 12 singleton lineages were identified, including 3 singleton taxonomic species, 3 misclassified named singletons, and 6 unnamed singleton lineages. Eight named species complexes contain >20 sequences, as indicated on the clade annotations in Fig. 3, and 186 unclassified sequences were assigned to phylogenetic clusters. An analysis of metadata associated with the ITS GenBank files, complemented with manually mined data from publications ($n = 341$, 75.44% of the whole data set), indicated that most (58.65%) *Naegleria* strains had been isolated from diverse freshwater sources, followed by 12.61% recovered from hot springs or other thermal waters. Human clinical isolates (cerebrospinal fluid, $n = 15$) correspond to the well-known pathogen *Naegleria fowleri*, while potential amphizoic fish pathogens recovered from gills and diverse internal organs correspond mostly to *N. americana* ($n = 9$) and *N. gruberi* ($n = 8$) strains. Association tests were highly significant for certain phylogroups and their geographic origin (Fig. 2C) ($P = 3.1308e$-14) or isolation source (Fig. 2D) ($P < 2.2e$-16). For example, *N. americana* is overrepresented among fish and soil isolates, while *N. fowleri* and *Naegleria australiensis* ITS sequences in GenBank have been overwhelmingly produced from clinical and hot spring isolates, respectively. The global sequence collection is dominated by isolates from East Asia (34.9%), particularly Japan ($n = 73$), followed by West Asia (17.6%) with 62 Iranian isolates, Europe ($n = 50$; 14. 66%), and North America ($n = 34$; 9.97%) with 24 Mexican isolates. Figure S1 online at https://doi.org/10.6084/m9.figshare.21200500 provides graphical overviews of these metadata, confirming that the eight best-sampled species have a very broad geographic distribution.

Based on the comprehensive ITS phylogeny presented in Fig. 3, we could classify 9 of the new Mexican *Naegleria* isolates to the species level. The 3HKMJS1 sequence was

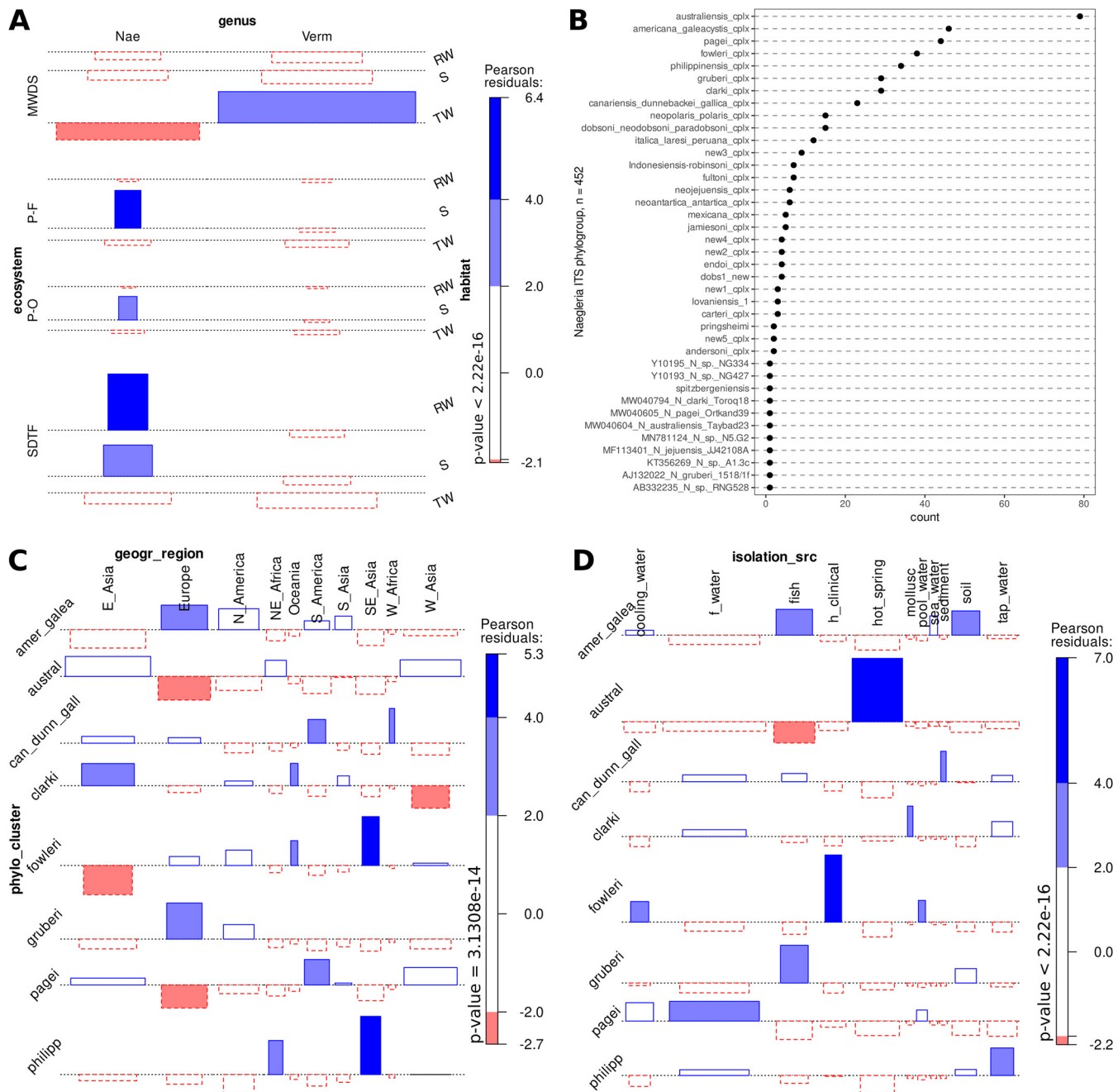

**FIG 2** Descriptive statistics and association analyses between the FLA taxa and ecological variables. (A) Three-way association between the genera of the Mexican isolates (Fig. 1, Table 2) generated in this study with their ecosystem and isolation habitat. (B) Cleveland dot plot showing the counts of *Naegleria* ITS sequences in each of the 40 lineages identified among the 452 subjected to the phylogenetic analysis (see Fig. 3). Two-way associations between *Naegleria* species identified by ITS sequences (Fig. 3) with the geographic region (C) or isolation source for the subset of 341 sequences with available metadata (D). The Chi-square Pearson residuals (A, C, and D) are displayed as positive or negative bars and colored according to significance (scale on the right-hand side; gray is nonsignificant), and their width is proportional to sample size. Here, we define the abbreviations used for the variables ecosystem (SDTF, seasonally dry tropical forest; P-O, mixed pine-oak forest; P-F, mixed pine-fir forest; MWDS, municipal water distribution system), genus (Aca, *Acanthamoeba*; Nae, *Naegleria*; Tet, *Tetramitus*; Ver, *Vermamoeba*), habitat (RW, river water; S, soil; TW, tap water), phylo_cluster (philipp, *Naegleria philippinensis*; austral, *N. australiensis*; can_dunn_gall, *N. canariensis*, *N. dunnebackei*, *N. gallica*; amer_galea, *N. americana*, *N. galeacystis*), and isolation source (f_water, freshwater; h_clinical, human_clinical).

identical to that of the *N. gruberi* reference strains EGB and NEG-M (35). The sequence of strain TA37-1 was identical to those of several Chinese, Japanese, and Iranian isolates, classified as *N. clarki*. However, their sequences are not identical to the type strain BG6, which forms a distinct sublineage within the *N. clarki* complex, together with the Mexican reference sequences CRET1 and CRET2, isolated from irrigation channels in

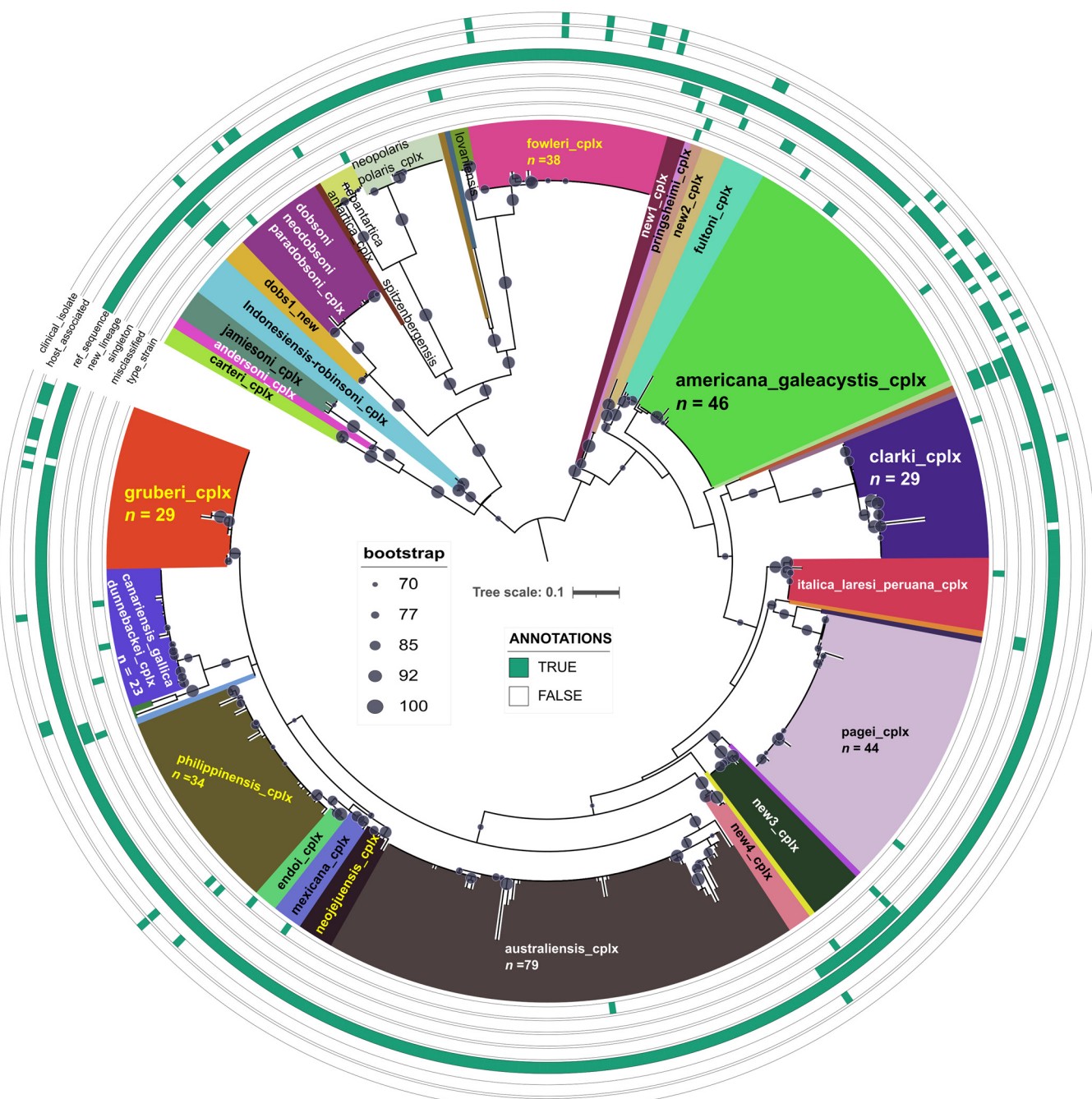

**FIG 3** Maximum likelihood phylogeny (ln*L* = −6649.305) estimated from 442 *Naegleria* sp. ITS sequences represent their global diversity in GenBank, as of August 2022. Well-supported clades of closely related species are colored and labeled as species complexes. The 10 Mexican *Naegleria* isolates cluster within the *N. americana*/*N. galeacystis*, *N. clarki*, and *N. gruberi* clades. Five new unnamed species complexes were found (new1 to new5) and multiple misclassified strains (see main text, annotations, and Table S3 online at https://doi.org/10.6084/m9.figshare.21200467). The alignment has 442 sequences with 577 columns, 361 distinct patterns, and 234 parsimony-informative, 67 singleton, and 276 constant sites. Best-fit model was TPM2u+F+G4, chosen according to BIC, out of 484 models analyzed. A compressed tar file with the tree and alignment is available online at https://doi.org/10.6084/m9.figshare.22011095.

Mexicali, Baja California (36). Consequently, TA37-1 was classified as *Naegleria* sp. within the *N. clarki* complex. The remaining 8 *Naegleria* isolates from our collection shared the same haplotype with CCAP1518/1G, the type strain of *N. americana* (34). Consequently, in line with the evidence from the 18S rDNA sequence analysis (Fig. 1), we tentatively classified the latter isolates as *N. americana*. Furthermore, these sequences resemble *N. galeacystis* isolate A.V.500 (X96578). The ITS1 and 5.8S regions of both species are identical (34), differing by a few indels in the ITS2 (37).

**TABLE 3** PCR primers used and designed in this study

| Primer[a] | Sequence (5′–3′) | Locus/target | Reference |
|---|---|---|---|
| F-566 | CAGCAGCCGCGGTAATTCC | 18S SSU rRNA/eukaryotes | 22 |
| R-1200 | CCCGTGTTGAGTCAAATTAAGC | 18S SSU rRNA/eukaryotes | 22 |
| JITS fw | GTCTTCGTAGGTGAACCTGC | ITS/Tetramitia | 23 |
| JITSccg_univ.F | GGTMTYCGTARGTGAACCTGC | ITS/Tetramitia+Amoebozoa | 23, This study |
| JITSccg_univ.R | CCSCTTAYTRATATGCTTAA | ITS/Tetramitia+Amoebozoa | 23, This study |
| ITSccg_univ.R | TTTTCCTCCSCTTAYTRATATGC | ITS/Tetramitia+Amoebozoa | This study |
| COI_NaeF208 | TGCTTTTCGTHGTRGTAATGCC | COI/Naegleria | This study |
| COI_NaeR873 | CWGAWGTATACATRTGATGAGC | COI/Naegleria | This study |
| COI_Amoebozoa.F | GTTCWACAAAYCAYAAAGAYATHGG | COI/Amoebozoa | 24, This study |
| COI_Amooebozoa.R | TAAACTTCVGGATGHCCAAAAAAYCA | COI/Amoebozoa | 24, This study |
| COI_Vermamoeba.F | GTTCWACAAACCATAAAGAYATWGG | COI/Vermamoeba | 24, This study |
| COI_Vermamoeba.R | TAAACTTCRGGATGMCCAAAAAAYCA | COI/Vermamoeba | 24, This study |
| COI_Tetramitia.F | TYACAACAAAYCATAAAGAATWGG | COI/Tetramitia | 24, This study |
| COI_Tetramitia.R | TAYACTTCTGGRTGTCCAAAAAACCA | COI/Tetramitia | 24, This study |

[a]The specific primers used for each amplicon generated in this study are listed in the corresponding GenBank entry.

An analysis of the ITS sequences generated for the two *Tetramitus* isolates recovered in this study (TS37-1 and TS37-2) (Table 2) suggests that they may correspond to a new species, at a *p*-distance of 0.032 substitutions/site (3.2% divergence) from *T. dokdoensis* (see Fig. S2 posted online at https://doi.org/10.6084/m9.figshare.21200527). For comparative purposes, *T. entericus* (AJ698856), a close relative of *T. dokdoensis*, presents a divergence of 2.17% from *T. dokdoensis*.

**Analysis of *Naegleria* COI sequences uncovers cryptic diversity within the *N. galeacystis*-*N. americana* complex and highlights multiple taxonomic inconsistencies.** To improve resolving power and species identification robustness, we adapted the popular "Folmer primers" (24) targeting mitochondrial cytochrome oxidase subunit I (COI) to match Vahlkampfiidae sequences (Table 3). A BLASTN search of the NCBI nonredundant (nr) nucleotide database retrieved 54 partial COI hits, deposited in GenBank as PopSet 1044963212. They were generated with primers reported by the Fulton and Wangh groups in their closed-tube barcoding paper (38). These sequences are shorter than those generated with our primers, which overlap the former region completely. After removal of 4 short sequences from the reference set, the final alignment had 396 sites (after trimming borders) and 60 sequences (54 haplotypes). No gaps were present in the matrix, and translation of the sequences revealed that they did not contain internal stop codons. Therefore, we concluded that they are *bona fide* COI coding sequences.

Figure 4 presents a Bayesian maximum clade credibility (MCC) phylogenetic estimate of the vahlkampfid COI sequences under a strict clock model, which unveils a significant amount of intraspecific (cryptic) diversity undetected in the SSU rDNA and ITS1-5.8S-ITS2 analyses described in previous sections. The Mexican *N. gruberi* isolate 3HKMJS1 grouped in very close proximity (range of 0.75% to 0.0% substitutions) to the three *N. gruberi* reference strains, being identical to that of strain NG560. This result strongly supports its classification as *N. gruberi*. Interestingly, four COI alleles were uncovered among eight isolates from our collection (*h* = 0.5) that share the same ITS sequence as the *N. americana* reference sequence (Fig. 3). These alleles formed a well-supported clade (0.99 posterior probability [PP]) with several reference strains, including the *N. americana* strains NG377 (Australian) and 5c1 but also with the *N. galeacystis* strain A.V.500 (ATCC 30294) (39). This strain was isolated in 1967 from soil samples collected in New York and was described as the novel genus and species *Adelphamoeba galeacystis* (40). It was later reclassified as *N. galeacystis* based on partial 18S rDNA sequence analysis (39). The Mexican XOCH1-3 and XOCH2-3 isolates were most closely related to *N. galeacystis*, differing by two residues from it (0.5%). The two reference *N. americana* sequences differed by 1 nucleotide and by 7 (1.77%) and 8 (2.2%) substitutions compared with *N. galeacystis*. Our TS3-2 and XOCH1-2 isolates shared the same COI haplotype, forming a unique branch within the *N. galeacystis*/*N. americana* clade at a distance of 3.53% to 2.79% substitutions from the corresponding reference sequences. Isolates TS3-1 and EAP1L shared the same haplotype,

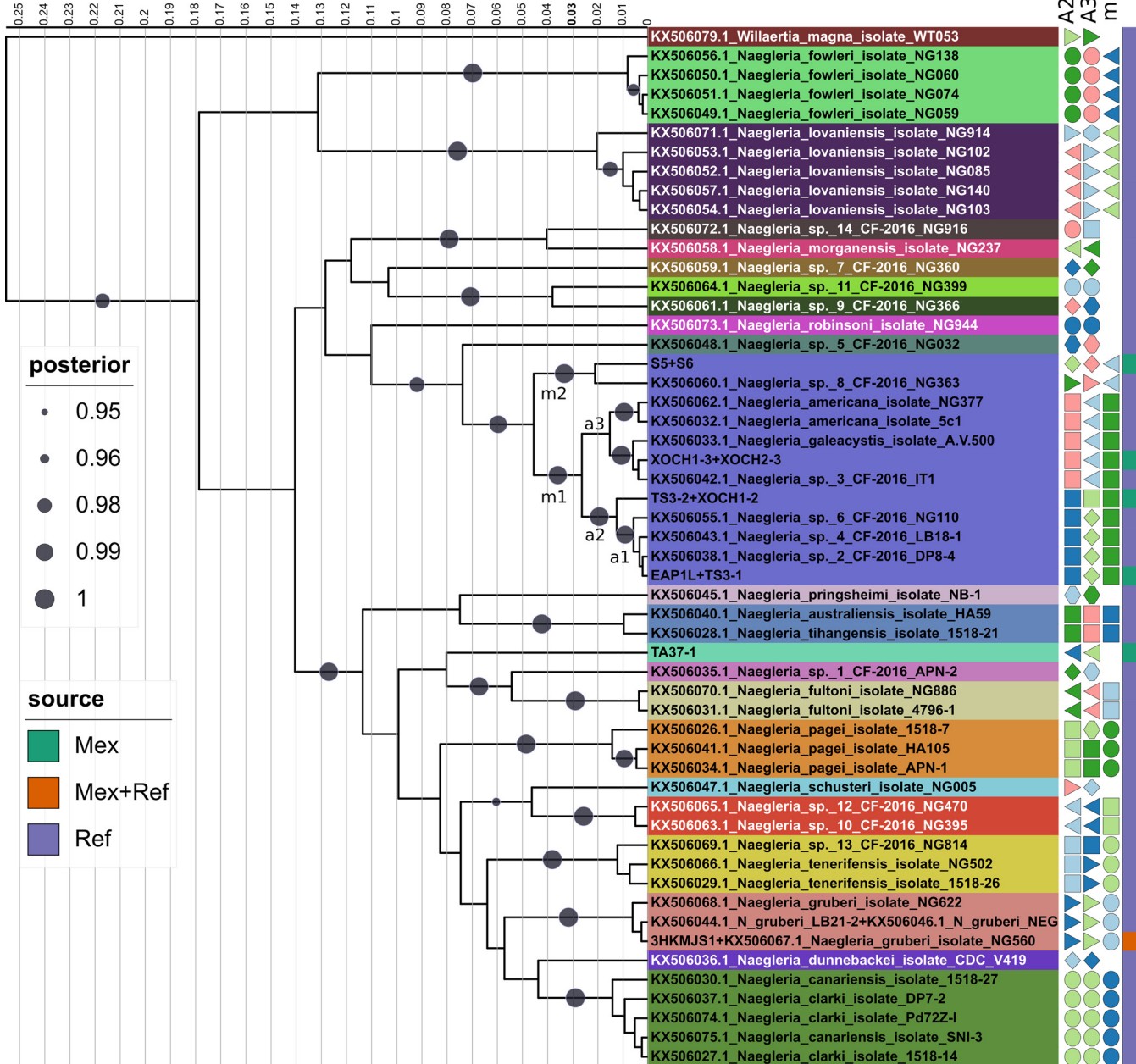

**FIG 4** Bayesian maximum clade credibility tree computed from 10,000 post-burn-in trees pooled from three independent BEAST 2 runs. Trees were estimated from *Naegleria* COI sequences under the TIMuf+G+I substitution model, using a strict molecular clock with mean rate of 1.0 and a Yule tree model with birth rate uniformly distributed (lower = 0.0 and upper = Infinity). ESS values for all parameters were ≫ 200 (range, 355 to 4,564). The alignment has 60 sequences (54 haplotypes) with 396 columns, 172 distinct patterns, 141 parsimony-informative sites, 31 singleton sites, and 224 constant sites. The sequence label color background corresponds to the most consilient species-like units (ASAP best partition), as defined in the main text. The columns of colored symbols correspond to the units delimited by the ASAP second and third partitions (A2 and A3) and the mPTP program (m). Note that mPTP units were not delimited for singleton species (see main text). The last column indicates the source of the sequences (Mex, this study; Ref, NCBI reference; Mex + Ref, both this study and NCBI reference). Bayesian branch support posterior probabilities are shown only for bipartitions with values of 0.95 to 1.00. Labeled ASAP (a) and mPTP (m) units within the *N. americana/N. galeacystis* clade correspond to those subjected to DNA polymorphism analyses (genetic differentiation and gene flow) summarized in Table 4. A compressed tar file with the tree and alignment is available online at https://doi .org/10.6084/m9.figshare.22011095.

with a divergence range of 3.79% to 3.05% from the *N. galeacystis* and *N. americana* reference sequences, respectively. Isolates S5 and S6, recovered from the pine-fir forest soil, shared the same haplotype, which was the most distant one. It grouped with *Naegleria* sp. 363 outside the main *N. galeacystis/N. americana* clade, which holds all previously mentioned Mexican and reference strains. This haplotype displays a divergence of 5.8% to 4.80% to 4.57% substitutions compared with the *N. galeacystis* and *N. americana*

reference sequences. These distances exceed the standard >2 to 3% interspecies divergence threshold used in many animal barcoding studies (21). Diverse barcoding studies on naked, testate, or scaled Amoebozoa have found intramorphospecies COI divergences that range from 0.6% to 1.7% in *Vannella* spp. (18), 0.2% to 0.9% in *Cochliopodium* spp. (41), and 0.13% to 2.5% in *Korotnevella* (42). Although intraspecific COI divergences are known to be lineage specific (43) and a clear "barcode gap" is often not found between morphospecies (44), including some Amoebozoa (18), our analysis revealed that the *N. americana* ITS haplotype hides a significant amount of cryptic diversity, with COI DNA sequence divergences ranging from 0.5% to 5.80%, compared with the reference sequences. Therefore, cryptic species (20, 44) may be present within this clade. Unfortunately, the reference COI sequences (38) have no associated metadata, limiting the ecological inferences and interpretation of the analysis, as their geographic origins and source habitats are undefined. Only the reference *Naegleria robinsoni* strain NG944 from the COI sequence set is shared with the global ITS set (Fig. 3).

**Evaluation of distance-based and phylogeny-aware single-locus species delimitation methods to define species-like units in the genus *Naegleria*, using COI sequences.** We used two recently published and contrasting single-locus species delimitation methods to gain further insight into species borders within the genus *Naegleria*. The "assemble species by automatic partitioning" (ASAP) algorithm is based on hierarchical clustering of genetic distances (45). In contrast, the phylogeny-aware "multirate Poisson tree processes" (mPTP) method (46) models speciation events by taking into account lineage-specific coalescence rates and a speciation parameter (see Materials and Methods). The delimitation results (called units here) of both algorithms are mapped on the ultrametric Bayesian COI phylogeny depicted in Fig. 4. The ASAP units were selected among the partitions with the first-, second-, and third-best ASAP scores, as suggested by Puillandre et al. (2020) (45). The three best-scoring ASAP partitions resolved 23, 27, and 31 units, with ASAP scores of 2, 6.5, and 6.5, respectively. ASAP units in the best partition were the most congruent with the 17 named (taxonomic) species clades (Fig. 4) and delimited 6 singleton units among the unclassified sequences. The largest unit clusters the *N. galeacystis* isolate A.V.500 together with the *N. americana* reference sequences and Mexican XOCH*, S*, TS*, and EAP1L isolates (Fig. 4). The second- and third-best ASAP partitions split the last clade into 5 and 6 units, respectively, both consistently grouping reference strains classified as *N. galeacystis* and *N. americana* with our XOCH* isolates. The latter two ASAP partitions also place the *Naegleria lovaniensis* isolate NG914 into a singleton unit. All ASAP partitions lump *Naegleria canariensis* with *N. clarki* into a single unit. However, the *N. clarki* COI sequences (38) are probably misclassified, as they group with *N. canariensis* and are close to *Naegleria dunnebackei* instead of the Mexican TA37-1 strain. The latter group clusters within the *N. clarki* complex in the 18S and ITS phylogenies (see Fig. 1 and 3).

It is important to note that the shifting point between speciation and coalescent processes cannot be determined for singleton species or units (those with a single haplotype sampled), as no coalescence events are available in the data set for that extant species (47). Therefore, we generated data sets without singleton species/units for mPTP-based delimitations. We defined them as taxonomic species with a single sequence or unclassified sequences consistently identified as singleton units by the three ASAP partitions. The overall support for the mPTP ML estimate of 11 units was very strong, as judged from the average support values (ASVs; 0.99295) over 10 independent Markov chain Monte Carlo (MCMC) runs (see Materials and Methods). The convergence of the 10 MCMC chains was satisfactory, given the negligible average standard deviation of delimitation support values (ASDDSVs; 0.000045). As shown in Fig. 4, except for the two units delimited by mPTP within the *N. americana*/*N. galeacystis* clade, the other units delimited by this program were perfectly consistent with those delimited by the ASAP best partition. These units are congruent primarily with currently accepted taxonomic species, based on the ITS standard by De Jonckheere and Brown (23). The best partition ASAP represents the most consilient delimitation compared with taxonomic species and the mPTP units. However, mPTP and all three ASAP partitions consistently grouped the *N. americana*/*N. galeacystis* taxonomic species pair as a single unit, suggesting that they may represent heterotypic

**TABLE 4** Genetic differentiation and gene flow estimates between selected ASAP (a)/mPTP (m) unit pairs of the *N. galeacystis/N. americana* and *V. vermiformis* species complexes

| Species(pop1-pop2)-locus no. of (sequences) (haplotypes) per unit[a] | Genetic differentiation | | | | Gene flow | |
|---|---|---|---|---|---|---|
| | Fixed differences[b] | Dxy[c] | $K_{ST}$[d] | P value[e] | $F_{ST}$[f] | Nm[g] |
| Ng(m1-m2)-COI (3, 13) (2, 10) | 9 | 0.05362 | 0.14194 | 0.0017** | 0.60835 | 0.32 |
| Ng(a1-a3)-COI (5, 6) (4, 5) | 9 | 0.03519 | 0.39155 | 0.0013** | 0.78657 | 0.14 |
| Ng(a2-a3)-COI (6, 7) (5, 6) | 6 | 0.03418 | 0.30503 | 0.0003*** | 0.68078 | 0.23 |
| Ng(a2-m2)-COI (3, 7) (2, 5) | 14 | 0.05712 | 0.34705 | 0.0089** | 0.74316 | 0.17 |
| Ng(a3-m2)-COI (3, 6) (2, 5) | 10 | 0.04937 | 0.32164 | 0.0080** | 0.70085 | 0.21 |
| Vv(m2-m3)-ITS (4, 14) (4, 11) | 49 | 0.11416 | 0.20265 | 0.0000*** | 0.82630 | 0.11 |
| Vv(m2-m4)-ITS (9, 14) (4, 11) | 41 | 0.10212 | 0.33698 | 0.0000*** | 0.83565 | 0.10 |
| Vv(m2-m7)-ITS (9, 14) (4, 11) | 48 | 0.11685 | 0.33718 | 0.0000*** | 0.84192 | 0.09 |
| Vv(m2-m1)-ITS (4, 14) (4, 11) | 70 | 0.15192 | 0.23353 | 0.0001*** | 0.86957 | 0.07 |
| Vv(m3-m4)-ITS (4, 9) (4) | 27 | 0.05653 | 0.39814 | 0.0002*** | 0.85199 | 0.11 |
| Vv(m3-m7)-ITS (4, 9) (4) | 22 | 0.05393 | 0.34644 | 0.0013** | 0.77222 | 0.15 |
| Vv(m3-m1)-ITS (4) (4) | 71 | 0.13275 | 0.39592 | 0.0109* | 0.90221 | 0.05 |
| Vv(m4-m7)-ITS (9) (4) | 21 | 0.05059 | 0.46015 | 0.0000*** | 0.83182 | 0.10 |
| Vv(m7-m6)-ITS (3, 9) (3, 4) | 29 | 0.07297 | 0.35072 | 0.0047** | 0.73485 | 0.18 |

[a]Unit definitions for *Naegleria* (Ng) and *Vermamoeba vermiformis* (Vv), by ASAP (a) and mPTP (m), as noted on Fig. 3 and 4.
[b]Number of fixed differences between populations.
[c]Average number of nucleotide substitutions per site between populations or lineages.
[d]Sequence-based statistic described by Hudson et al. (15).
[e]Probability obtained by the permutation test (15) with 1,000 replicates. *, $0.05 \geq P > 0.01$; **, $0.01 \geq P > 0.001$; ***, $P \leq 0.001$.
[f]Sequence-based estimate described by Hudson et al. (15).
[g]Effective number of migrants.

synonyms. Based on these results, we determined that a TPM2u+G+I distance of 0.03 is a conservative and consilient cutoff to delimit *Naegleria* species on the ultrametric Bayesian MCC tree presented in Fig. 4.

**Population genetic structure, genetic differentiation, and gene flow in the *N. galeacystis/N. americana* species complex.** In order to challenge the ASAP/mPTP delimitations within the *N. galeacystis/N. americana* clade described in the previous section, we computed Hudson's $K_{ST}$* index of population genetic differentiation (48) between selected units. There is a highly significant population subdivision ($K_{ST}$*, 0.47121; $P < 0.0001$) when considering the entire set of 16 *N. galeacystis* and *N. americana* sequences, either when assuming 5 units delimited by the second-best ASAP partition or the 2 units delimited by mPTP ($K_{ST}$*, 0.14194; $P = 0.0012$). This finding is consistent with the very high fixation index ($F_{ST}$, 0.91362) and negligible estimate of migrants between subpopulations ($Nm$, 0.05). Table 4 summarizes the pairwise genetic differentiation and gene flow estimates between selected ASAP and mPTP units. All pairs have multiple fixed nucleotide differences (i.e., private polymorphisms) (range, 6 to 19) and mean subpopulation differences ($D_{xy}$) in the range of 1.9% to 5.5%. Despite the small sample sizes, all pairwise $K_{ST}$* genetic differentiation indices are significant. The high $F_{ST}$ fixation indices (range, 0.79 to 0.9) further denote strong lineage differentiation. This finding is consistent with the negligible numbers of effective migrants per generation estimated ($Nm$ range, 0.06 to 0.14), suggesting minimal genetic flux between these units. These results are consistent with the second-best scoring ASAP delimitations and provide additional and independent evidence for at least three significantly differentiated lineages within the *N. americana-N. galeacystis* species complex. In conclusion, these tree units may represent novel species.

**Phylogenetic analysis of ITS1-5.8S-ITS2 sequences uncovers multiple cryptic lineages within *Vermamoeba vermiformis*, grouped by ASAP in at least seven species-like units, and identifies many misclassified sequences in public databases.** We optimized primers to target *V. vermiformis* (Table 3), formerly *H. vermiformis* (31), as none have been published previously for this species. However, diverse primers, particularly the JITS set by De Jonckheere and Brown (23), which was designed to amplify the ITS region from vahlkampfids, have been used to amplify *Vermamoeba* (*Hartmannella*) sequences, often unknowingly. This amplification has led to the accumulation of ITS sequences from isolates and cloned environmental amplicons misclassified as belonging to diverse vahlkampfids (supergroup Discoba), which correspond to *Vermamoeba* (supergroup Discosea). Morpho-

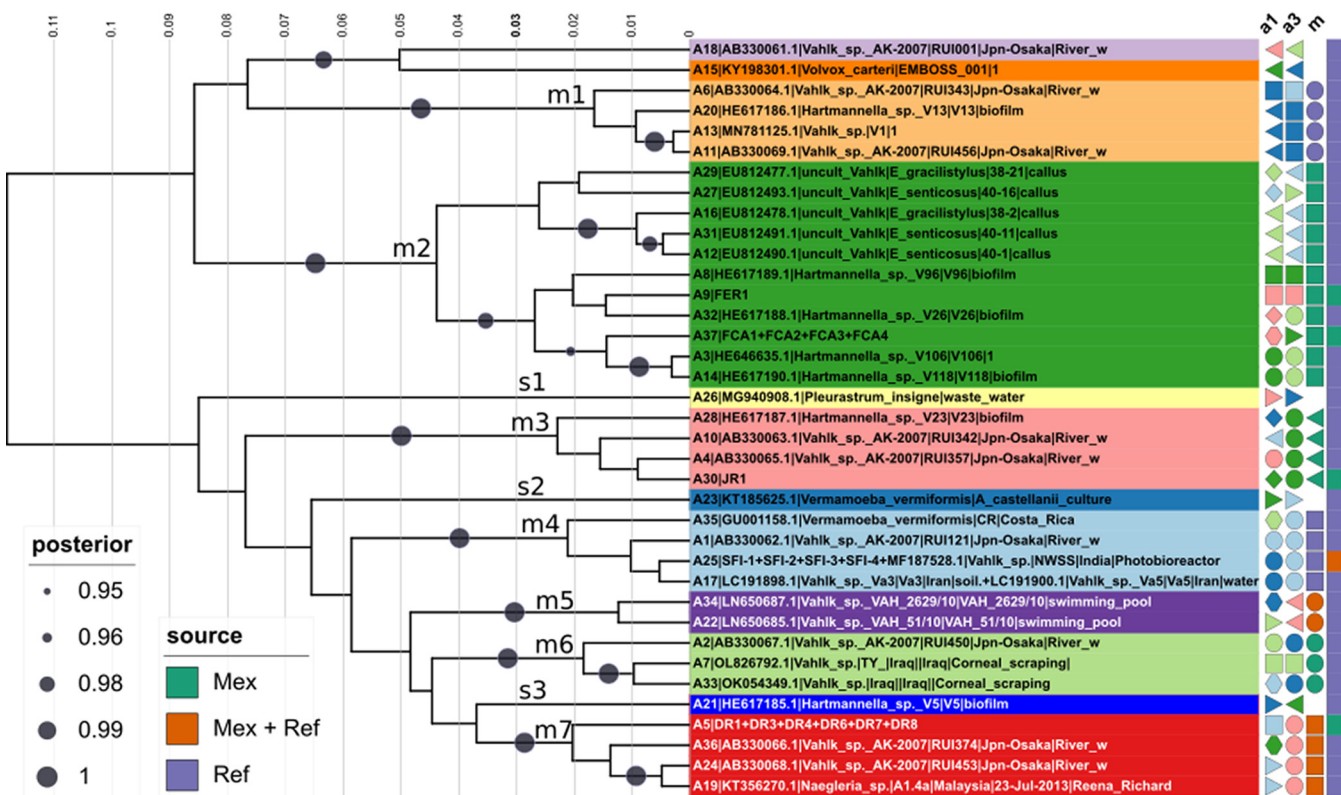

**FIG 5** Bayesian maximum clade credibility tree computed from 10,000 post-burn-in trees pooled from three independent BEAST 2 runs. Trees were estimated from *Vermamoeba* ITS sequences under the best-fitting TPM2u+G substitution model, using an optimized relaxed clock (ORC) with mean rate of 1.0 and a Yule tree model with birth rate uniformly distributed (lower = 0.0 and upper = Infinity). ESS values for all parameters were ≫ 200 (range, 1,434 to 5,493). The alignment had 50 sequences (37 haplotypes; first number in label) with 624 columns, 221 distinct patterns, 157 parsimony-informative sites, 55 singleton sites, and 412 constant sites. Tree annotations are as in Fig. 4. Labeled mPTP (m) units correspond to those subjected to DNA polymorphism analyses (genetic differentiation and gene flow) summarized in Table 4. Colored stripes group sequences into 12 conservatively defined units. A compressed tar file with the tree and alignment is available online at https://doi.org/10.6084/m9.figshare.22011095.

logical and sequence analyses demonstrated this finding in previous studies (49, 50). Here, we update the list of misclassified *Vermamoeba* ITS sequences accumulated in the NCBI nucleotide database (Table S2 online at https://doi.org/10.6084/m9.figshare.21200491). Using the ITS sequence from *V. vermiformis* SFI-1 as the bait, only 12 out of 53 BLASTN hits found in the NCIB nr database with a query coverage (qcov) of ≥40%, an identity of >76.7%, and an E value of <1e-71 are correctly classified as *V. vermiformis* (or *H. vermiformis*). We found that 43% hits are misclassified as *Vahlkampfia*; 1.9% each as *Chroococcus* (Cyanobacteria), *Naeglaria*, *Pleurastrum* (Chlorophyta), or *Volvox* (Chlorophyta); and 13.2% belonging to cloned amplicons from uncultured organisms classified as Vahlkampfiidae. We selected a subset of 34 sequences with qcov of ≥89% and sequence identity of >80% for further analysis. They included sequences of FLA recovered from rivers (RUI labels) in Osaka, Japan (51), and ITS amplicons cloned from environmental DNA purified from plant tissue cultures in Germany (52), misclassified as *Naeglaria* sp., as reported previously (49, 50). We found additional misclassified sequences from drinking water treatment plants in Malaysia (53), water sources in Iran (54), corneal scrapings from Iraq, *Chlorella* cultures from India (55), and public baths in Hungary (56).

Figure 5 presents the Bayesian MCC tree estimated using an optimized relaxed clock (ORC) model (57), as we found strong evidence (Bayes factor, 32.19888; see Materials and Methods) in favor of this model over a strict molecular clock. The phylogeny resolved seven deeply branching and well-supported clades with ≥0.99 PP (labeled as C1 to C7) and three singleton lineages (S1 to S3). Trophozoite morphotypes of the reference isolate V13 from C2, growing in liquid and on agar media, were shown by De Jonckheere et al. (49) to correspond to those of *V. vermiformis*. The 16 Mexican isolates grouped in the perfectly

supported clades m2, m3, m4, and m7, along with reference sequences from various continents. The SFI-* isolates all shared the same haplotype with that of the Indian *Chlorella* culture isolate NWSS (MF187528) (55), forming the strongly supported (0.99 PP) clade m4 together with the Costa Rican *V. vermiformis* CR reference isolate (58), the Japanese RUI121 isolate (51), and other isolates from Iran (54). The Mexican DR* sequences from showerhead biofilm isolates formed a distinct lineage within clade C7, along with two Japanese sequences (RUI374 and RUI453) (51) and the Malaysian sequence KT356270.1 (53). The Mexican FCA* isolates recovered from the activated charcoal compartment of a domestic water filtration unit shared the same and distinct ITS sequence, being most closely related to the two reference sequences V106 and V118 from Ghana (49). These sequences are part of the strongly supported (0.99 PP) and deeply branching clade m2, along with cloned *Vermamoeba*-related amplicon sequences recovered from *Eleutherococcus* sp. tissue cultures (52) and that of the Mexican isolate FER1. The Mexican showerhead biofilm isolate JR1 was associated with a perfectly supported clade containing Japanese river water RUI isolates (51) and the African (Ghana) biofilm strain V23 (49).

The ASAP algorithm (45) uncovered 30, 12, and 19 units in the first, second, and third best-scoring partitions found among the 50 sequences (37 haplotypes) analyzed. The respective unit delimitations are shown on the phylogeny (Fig. 5) along with other ecological data, such as isolation source and country, revealing that many of these molecular species-like units have a broad geographic distribution spanning multiple continents. Furthermore, the units delimited by ASAP's second best-scoring partition (excluding singletons) were highly consistent with the seven strongly supported units (m1 to m7) delimited by mPTP (ASV, 0.935908; ASDDSV, 0.000070), and all of them formed monophyletic groups with perfect (PP, 1) support. Therefore, based on a consilience criterion, there was substantial evidence for at least seven species-like units in the ITS data set (Fig. 5). This figure rises to at least 12 units when singletons are taken into account, as denoted by the tree-label color shadings.

**Analysis of *V. vermiformis* ITS sequence polymorphisms reveals highly significant population subdivision.** We performed diverse DNA polymorphism sequence analyses to provide independent evidence for the evolutionary significance of the ITS lineages delimited within *V. vermiformis* using ASAP and mPTP. Table 4 shows compelling evidence for population subdivision found in the *V. vermiformis* ITS data set. Fixed differences between pairwise comparisons of the three main phylogenetic lineages ranged from 21 to 71, and the permutation tests for the Hudson's $K_{ST}$ statistic of population subdivision (48) were all significant. Six out of nine pairwise comparisons were very highly significant (Table 4). These results, along with the very high fixation indices (range, 0.73 to 0.90), and relatively low estimates of migrants (range, 0.05 to 0.18), strongly reinforce the notion that the units revealed by ASAP and mPTP represent independently evolving lineages (Fig. 5), consistent with species-like units (Table 4).

***Vermamoeba vermiformis* COI sequences from Mexican isolates congruently resolve the same three lineages revealed by ITS.** Figure 6A shows the maximum likelihood phylogeny estimated from the *V. vermiformis* complex COI sequences, rooted with two *Acanthamoeba* sequences retrieved from the two complete mitochondrial genomes available for the genus (NC_001637 and KP054475.2). In addition to the *V. vermiformis* reference COI sequences from two complete mitochondrial genomes (KT185627 and NC_013986), we found an additional mitochondrial genome assembly without annotation (ASM2069642v1), corresponding to isolate TW EDP1, recovered in 2014 from a drinking water faucet in Ivry sur Seine, France (59). As detailed in the Materials and Methods, we used tblastn to identify the scaffold and DNA region coding for the *cox1* gene. The same three lineages of Mexican sequences (FCA*, SFI*, and DR*) found in the ITS phylogeny (Fig. 5) were recovered in the COI tree (Fig. 6A). The COI sequences did not uncover any further diversity within the three Mexican lineages, suggesting that they are highly clonal (60). We could not obtain COI amplicons for the isolates FER1 and JR1. The SFI* haplotype was most closely related to that of the French isolate TW EDP1 (59) but was also very similar to that of the reference sequence HaveOM_p35 from *Vermamoeba* (*Hartmanella*) *vermiformis* ATCC 50236 (61). The third-best ASAP partition was the most consilient with the results obtained from the ITS data and suggests the

A

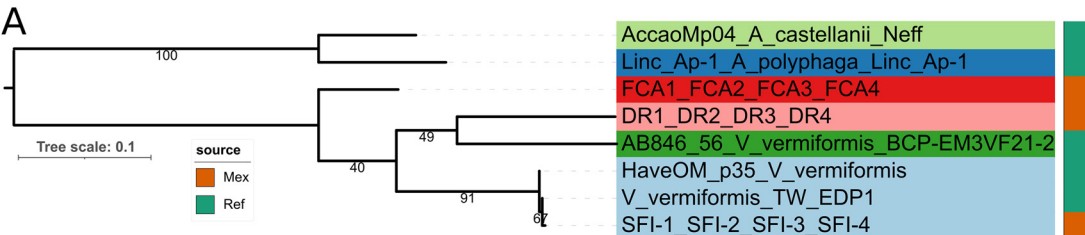

B

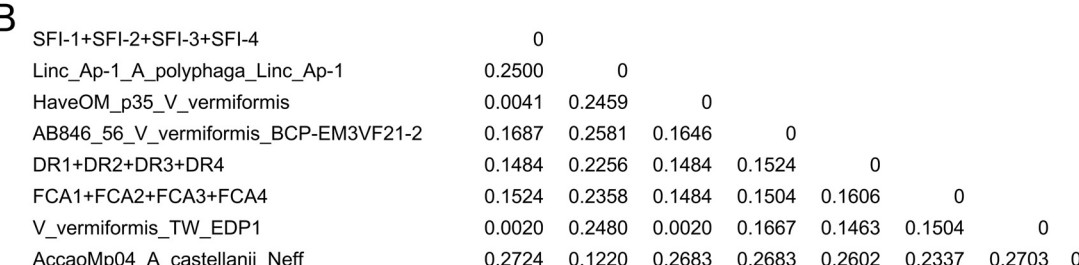

|  | | | | | | | |
|---|---|---|---|---|---|---|---|
| SFI-1+SFI-2+SFI-3+SFI-4 | 0 | | | | | | |
| Linc_Ap-1_A_polyphaga_Linc_Ap-1 | 0.2500 | 0 | | | | | |
| HaveOM_p35_V_vermiformis | 0.0041 | 0.2459 | 0 | | | | |
| AB846_56_V_vermiformis_BCP-EM3VF21-2 | 0.1687 | 0.2581 | 0.1646 | 0 | | | |
| DR1+DR2+DR3+DR4 | 0.1484 | 0.2256 | 0.1484 | 0.1524 | 0 | | |
| FCA1+FCA2+FCA3+FCA4 | 0.1524 | 0.2358 | 0.1484 | 0.1504 | 0.1606 | 0 | |
| V_vermiformis_TW_EDP1 | 0.0020 | 0.2480 | 0.0020 | 0.1667 | 0.1463 | 0.1504 | 0 |
| AccaoMp04_A_castellanii_Neff | 0.2724 | 0.1220 | 0.2683 | 0.2683 | 0.2602 | 0.2337 | 0.2703 | 0 |

**FIG 6** (A) Maximum likelihood phylogeny (ln*L* = −1352.024) of Vermamoeba COI sequences found in an IQTree2 search under the best-fitting K3Pu+F+I model selected using the Bayesian information criterion (BIC) out of 286 DNA models. The alignment has 13 *Vermamoeba* plus 2 *Acanthamoeba* sequences, collapsed to 8 haplotypes, with 492 columns, 132 distinct patterns, 128 parsimony-informative sites, 57 singleton sites, and 307 constant sites. The bar indicates the number of expected substitutions per site under the selected model. Tree annotations are as in Fig. 4. Numbers on internal branches correspond to bootstrap support values. (B) Matrix showing the number of pairwise base differences per site (*p*-distances) between the eight COI haplotypes. A compressed tar file with the tree and alignment is available online at https://doi.org/10.6084/m9.figshare.22011095.

existence of four units among the *Vermamoeba* COI sequence set, as follows: one holding the Mexican FCA* isolates; the second containing only the reference isolate BCP-EM3VF21-2 (50); the third one corresponding to the DR* isolates; and the fourth and largest unit containing the tightly clustered ATCC 50236 (61), TW ED1 (59), and the Mexican SFI* isolates (divergence, <0.4%). Strain ATCC 50236 is an axenic derivative of ATCC 30966 (62), isolated in 1964 by F. C. Page from an unspecified environmental freshwater source in the United States (https://www.atcc.org/products/30966). Strain BCP-EM3VF21-2 was isolated from the culture of *Xerochlorella olmae*, a green algal species (Trebouxiophyceae, Chlorophyta). However, its locality of origin is uncertain, as it may have been associated with the algal sample at its type locality (desert soil crust, Mojave Nature Preserve, CA), or it may be a postcollection contaminant (50). The small distances between isolates from the SFI* unit are in stark contrast with those between units. As shown in Fig. 6B, *p*-distances between the four *Vermamoeba* units are well over 0.14 (14% divergence), providing overwhelming support for our claim that they represent different species.

In summary, both the ITS and COI markers for *V. vermiformis* have a much higher resolving power than the widely used 18S rRNA gene (17), as reported previously by Fučíková and Lahr (50). Both markers provide compelling and congruent evidence for the existence of multiple species within the taxonomic species *V. vermiformis*.

**Agreement between morphotypes and molecular classification of representative isolates of each genus of FLA identified.** We used phase-contrast light microscopy to determine the morphotypes of representative isolates of each genus to validate their molecular classifications (13, 31, 63). The results are summarized in Fig. 7. *Acanthamoeba* trophozoites of two genotype T13 isolates are naked and flattened, with prominent, short, sharp, hyaline subpseudopodia, which are flexible and tapering to a fine tip (acanthopodia), characteristic of the acanthopodial morphotype (64). Cells lack an adhesive uroid (Fig. 7A and B). Cysts are double walled, with polygonal endocysts (Fig. 7B).

*Vermamoeba vermiformis* complex trophozoites display a typical monotactic morphotype characterized by elongated, smooth, and subcylindrical cells with a >6× length/width ratio. They produce smooth lobose pseudopodia (lobopodia), displaying monoaxial flow of the cytoplasm. When changing direction, the trophozoites often branch (Fig. 7C), becoming polypodial for a short time, quickly reverting to the monopodial form. Some display a basal uroid (Fig. 7D).

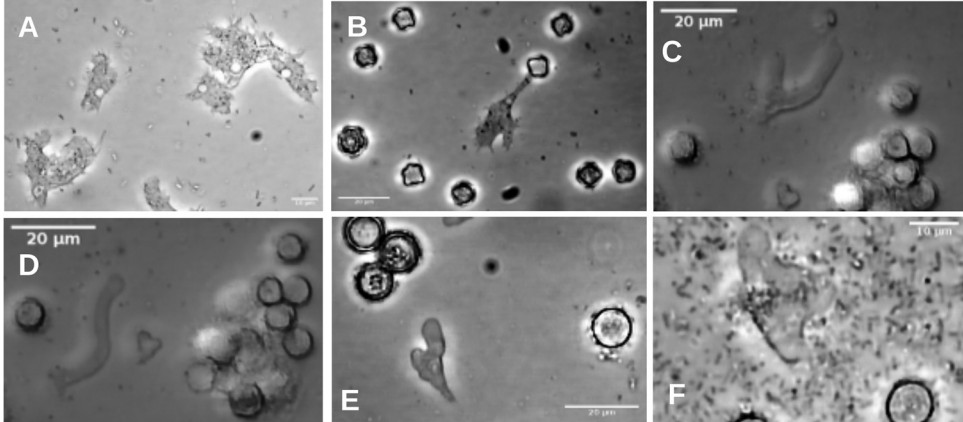

**FIG 7** Phase-contrast micrographs of selected isolates from each genus recovered in this work reveal that their morphotypes are consistent with their sequence-based classification. (A) *Acanthamoeba* sp. 4MJS2 (genotype T13) trophozoites with large contractile vacuoles and acanthopodia. (B) *Acanthamoeba* T13 5HKMJS2 with double-walled, polygonal cysts and a trophozoite with acanthopodia. (C) *Vermamoeba vermiformis* SFI-1 cysts and a branching trophozoite while changing direction and hyaline anterior caps. (D) *Vermamoeba vermiformis* SFI-1 cysts and a trophozoite with a basal bulbous uroid, with a >6× length/width proportion. (E) *Naegleria americana* TS3-1 immature (thin-walled) and mature (thick-walled) cysts and a trophozoite displaying characteristic cytoplasmic eruptions on the advancement front and an uroid on the trailing end. (F) *Tetramitus* sp. TS37-1 trophozoite and cyst.

The two vahlkampfid isolates shown (Fig. 7E and F) display characteristic eruptive cytoplasmic flows, seen as large hyaline projections on the anterior end of the locomotive form, which define the "eruptive morphotype." The trophozoites have a variable, although mostly a monopodial shape. A well-developed uroid is visible in many *N. galeacystis*/*N. americana* complex trophozoites, often terminating in several filaments, as first observed by Napolitano and colleagues (40). Flagellated cells were seen but could not be photographed due to their rapid swimming in living cultures.

## DISCUSSION

In this study, we performed a prospective survey of the molecular diversity of free-living amoebae across the representative ecosystems of the State of Morelos (Central Mexico). This regional-scale diversity was analyzed in the context of comprehensive sets of nuclear (18S rDNA and ITS) and mitochondrial (COI) sequences found in GenBank from isolates collected worldwide. In Mexico, studies using molecular sequence data to analyze the diversity of FLA have focused on amoebae of clinical importance like *Acanthamoeba* (65), *Balamuthia mandrilaris* (66), *Naegleria* (36, 67), and *Vermamoeba vermiformis* (67). Most of them used genus- or species-specific primers targeting the 18S rRNA or ITS regions, biasing and limiting the assessment of the diversity of FLA in the sampled habitats. On the other hand, studies with a broader ecological scope in natural (68, 69) and urban ecosystems (70, 71) have relied on morphological characterization for taxon identification. However, it is well established that morphological characters, even for testate amoebae like the well-studied Arcellinida, are often misleading for reliable taxonomic identification of FLA due to their homoplasious nature (72, 73). Furthermore, morphospecies typically hide multiple cryptic species (14, 18, 74).

Our study is the first one to survey the diversity of FLA in Mexico on a regional scale using state-of-the-art evolutionary genetic analyses of nuclear and mitochondrial loci coupled with formal statistical analyses to uncover associations between evolutionary lineages and ecological variables. As suggested by the CBOL protist working group (14), we set up a two-step strategy, starting the molecular characterization of our complete collection with universal eukaryotic primers spanning the V4 and V5 regions (22). In our hands, the F-566/R-1200 oligonucleotides performed very well as PCR and sequencing primers, demonstrating their suitability to analyze a diverse collection of Amoebozoa (*Acanthamoeba* and *Vermamoeba*) and Tetramitia (*Tetramitus* and *Naegleria*). To the best of our knowledge, these primers have not

been used previously to analyze the diversity of isolates of FLA. Their value for broad-scope diversity and ecological studies of FLA is underlined by the results of the association analysis between our isolates' 18S rDNA-based taxonomic identification and broad ecological variables. That analysis revealed a tight association of *V. vermiformis* with the urban water distribution system in Cuernavaca. In contrast, the *Naegleria* isolates had a broad distribution in water and soil along the different natural ecosystems sampled (Table 1). Our results are in line with several previous studies that analyzed the diversity of FLA in rivers (51), rural borehole water (75), drinking-water processing plants (53, 76), domestic and clinical tap water distribution systems (54, 77–79), public baths (56), and swimming pools (80, 81). These publications report a high prevalence of *V. vermiformis* in these aquatic, mostly human-made ecosystems. In addition, a few landmark papers have identified correlations between environmental variables and 18S-amplicon-based diversity of FLA in drinking water biofilms. For example, *Vermamoeba*-like 18S rDNA sequences were abundant in biofilms developed on unplasticized polyvinyl chloride and copper plumbing (82), and their relative abundance increases with temperature (83). Another study reported that DWDS biofilms are mostly colonized by a single active species of FLA (84); *Naegleria* species dominating over *Vermamoeba* in biofilms with higher bacterial richness; and the presence of indicator eukaryotic lineages, such as Nematoda and Rotifera (85). Although our study did not evaluate these biotic and abiotic variables, they may explain the striking dominance of *Vermamoeba* in our DWDS samples from the showerhead and filtration unit biofilms (Table 1).

In a second step, we optimized existing primer formulations to target the nuclear internal transcribed spacer region (ITS1-5.8S rRNA-ITS2) (23) and the mitochondrial cytochrome oxidase subunit I (COI or *cox*1) (24) of Amoebozoa and Tetramitia (Table 3). The ITS region is the second most frequently used marker to study the diversity of FLA (14). The popular JITS primers (23, 34) have been instrumental for constructing the current molecular standard of species identification and description for vahlkampfiids. They bind to highly conserved regions in the 5′ and 3′ ends of the 18S and 28S rRNA genes, respectively. Although designed for Vahlkampfiidae, they have been used to amplify the ITS region from distantly related Amoebozoa, like *Vermamoeba* (49, 50). Based on this empirical evidence, we reasoned that the regions targeted by the JITS primers are ideal for developing improved and standardized ITS barcoding primers for Amoebozoa. The ITS sequences of our *V. vermiformis* isolates allowed us to gain taxonomic and ecological insights into the cryptic diversity found in this species. The power of this analysis was significantly enhanced thanks to the accumulation in the public sequence databases of *V. vermiformis* ITS sequences generated by different groups with the JITS primers from contrasting habitats around the World. As noted previously, many of these sequences are misclassified as *Naegleria* or vahlkampfids (49, 50). We could therefore analyze our 16 *Vermamoeba* isolates (5 haplotypes) in the context of 34 carefully selected reference sequences originating from different continents. We uncovered 37 haplotypes ($h = 0.74$) grouped in strongly differentiated and supported phylogenetic clades. We applied the recently published distance-based (ASAP) (45) and the phylogeny-aware mPTP (47) single-locus species-delimitation algorithms, combined with diverse population genetic statistics, to identify evolutionary significant units (ESUs) of diversity (86) in our sample. ESUs are akin to evolutionary species-like units (87). We found strong and congruent evidence from these independent approaches for at least seven ESUs, most likely representing cryptic species lumped in *V. vermiformis*. However, we found that both delimitation methods require some data preprocessing to perform optimally, like collapsing sequences to haplotypes. In addition, mPTP requires at least two haplotypes per ESU, as the shifting point between speciation and coalescent processes cannot be determined for singleton species or units because no coalescence events are available for them in the data set (45). Therefore, we paid great attention to identifying and excluding singleton units from data sets to be analyzed with mPTP to avoid lumping these lineages with sister units or clades. In addition, we found that mPTP performed much better when provided with ultrametric Bayesian trees computed under a strict clock model compared

with maximum likelihood trees (data not shown). This finding is consistent with the conclusions of a recent and comprehensive study that compared the performance of several single-locus species delimitation programs on a species-rich beetle COI data set (46). After these issues were properly handled, mPTP provided delimitations consistent with the most conservative ASAP proposals. However, the mPTP program has the advantage of inferring a single set of units, of which the support can be computed in a Bayesian framework, while ASAP provides an arbitrary number of unit sets (partitions), and the user has to decide which one is the most reasonable.

It is well established that single-locus species-delimitation methods are limited in making correct delineations (44, 87–89). They are affected by diverse intrinsic factors, such as species distribution range, effective population size, demographic history, speciation rate, and species age, and by extrinsic factors, such as sampling effort (46, 89, 90). However, when methods operating on completely unrelated criteria congruently delimit the same units, there is a high likelihood that they are significant and possibly correct (89). Based on various criteria of optimal sample size for species delimitation (90), the size of our ITS data set ($n = 50$) is adequate to uncover potential cryptic species within *V. vermiformis* for the distance, phylogeny-aware, and population genetics methods employed here. The possibility that the delimited units may correspond to *bona fide* species, conceived as genetically coherent and evolutionary independent lineages, is strongly supported by three additional lines of evidence. First, the three largest units contain isolates and haplotypes from different continents, meaning that although by no means thorough, a global sample of haplotypes from these units was analyzed, making the inference of their borders robust (46, 90). Second, population genetic structure analyses revealed that these units are significantly differentiated, with negligible gene flow across them. Moreover, as our Mexican tap water isolates highlighted, such differentiated units coexist locally, as expected for well-differentiated microbial species (87). Third, the unlinked COI sequences congruently separate the Mexican *Vermamoeba* isolates in three deeply divergent phylogenetic lineages, delimited as independent units by ASAP. Unfortunately, only three *Vermamoeba* reference COI sequences could be found in the sequence databases, limiting the ecological, taxonomic, and evolutionary inferences that could be made based on this locus. However, the evidence gained from both loci strongly suggests that *V. vermiformis*, currently considered a monospecific genus (17, 31), contains multiple cryptic species and should be treated as a species complex in need of a thorough taxonomic revision. We refrain from making taxonomic proposals until a larger collection of ITS, COI, and additional sequences from globally sampled *V. vermiformis* complex strains becomes available. As current molecular systematics standards dictate, evidence from multilocus sequence analyses of unlinked loci sampled from the same set of strains should be obtained before making taxonomic proposals (72, 87, 91, 92). We expect that the optimized ITS and COI primers developed in this study for *Vermamoeba* will foster interest in studying the biogeography and ecology of this globally distributed genus (17) of great relevance in clinical and drinking water distribution systems (76–78, 82, 83, 91). These primers could also be helpful in the clinical setting, given the increasing number of reports from around the globe indicating that *V. vermiformis* is the second most prevalent species of the FLA recovered from human keratitis after *Acanthamoeba castellanii* (93). Sequence analysis of amplicons generated with these primers could determine whether only some of the cryptic intraspecific lineages of the *V. vermiformis* species complex are of clinical concern, which should have the highest priority for much-needed comparative genome sequencing projects.

The second largest group of isolates recovered in this study was Vahlkampfiidae. We classified the isolates in this group at the species level using the currently accepted ITS sequence standard (23, 34). The present work analyzed a comprehensive selection of the *Naegleria* ITS sequences available in GenBank, revealing the global phylogenetic diversity of the genus (Fig. 3). All sequences from our isolates and 187 (95%) unclassified database sequences could be assigned to 1 of the phylogroups resolved. Furthermore, mining

metadata associated with the reference sequences allowed us to test diverse ecological hypotheses. We uncovered highly significant associations between some of the largest phylogroups with their geographic origin and isolation source (Fig. 2). However, these results must be carefully interpreted, given the small sample sizes and uneven sampling of habitats and ecosystems across the globe. In addition, the very coarse isolation source categories defined necessarily miss potential finer-grained habitat specialization features (64). Some 18S-amplicon-based ecogenomic studies have identified potential associations between *Naegleria* species and environmental or biotic factors. For example, several independent studies have consistently found that the broadly distributed, thermophilic, and pathogenic species *N. fowleri* is more prevalent in biofilms at high temperatures, bacterial richness, and densities (84, 94). However, the relationships between these and other variables are complex and often inconsistent (84).

A primary goal of this study was to challenge the ITS-based taxonomy of the genus *Naegleria* using COI sequences. We adapted the widely used Folmer COI primers (24) to barcode *Naegleria*. An analysis of the *cox1* sequences from our *Naegleria* isolates revealed a significant amount of cryptic diversity, with a high haplotype diversity ($h = 0.5$) hidden under the single ITS haplotype shared by the *N. americana* isolates. We could significantly expand the phylogenetic, population genetic, and taxonomic analysis of our *Naegleria* COI sequences thanks to the 54 reference sequences found in GenBank (38). However, these sequences were not analyzed in detail. Our study is the first work on *Naegleria* diversity that makes use of this overlooked data set for molecular systematics and ecology of the genus. Of particular relevance for our study were two reference sequences from *N. americana*, the sequence of *N. galeacystis* A.V.500, and five additional sequences of unclassified *Naegleria* isolates. They grouped in a well-supported clade along with the Mexican COI sequences classified as *N. americana* ($n = 8$) based on their ITS sequences. An analysis of this small ($n = 16$) but still significant (90) set of *N. americana*-like isolates using ASAP (45) and mPTP (47) consistently placed the *N. galeacystis* and *N. americana* isolates in a single unit. This result suggests that *N. americana* (34) could be a heterotypic synonym of *N. galeacystis* (39). According to the fundamental Priority rule (Principle III) of the International Code of Nomenclature (95), the last name would have priority because it was published first (39). ASAP delimited four additional units within the *N. americana*/*N. galeacystis* clade. The highly significant population genetic differentiation statistic $K_{ST}^*$ (48), substantial pairwise $F_{ST}$ fixation indices between the units, and negligible gene flow (96) across them reinforced the notion that at least three ESUs exist in the *N. galeacystis*/*N. americana* COI clade. Each of them has Mexican representatives, which implies that multiple genetically independent but closely related units coexist in space and time, further underlying the notion that they may correspond to *bona fide* cryptic species (87). This analysis revealed that the analyzed COI segment is significantly more polymorphic than the ITS region and, therefore, better suited for ecological inference and molecular systematics studies. Furthermore, COI markers pave the way for finer-grained association studies between FLA lineages and environmental variables, particularly if coupled with high-throughput sequencing of lineage-specific COI barcodes generated directly from environmental DNA, as reported recently (74).

The COI region has the following additional potential advantages over the ITS region: it is haploid, lacks introns, and experiences no or minimal recombination and the impact of insertions or deletions (indels) is negligible when comparing closely related species (21). Furthermore, *cox1* genes typically have a much lower intrastrain sequence variability than the ITS loci, as demonstrated by sequencing multiple cloned amplicons of these loci from the same strain (18). Diverse population genetics methods require the ploidy of the analyzed loci to be known for optimal performance. However, the ploidy of most species of FLA is currently unknown due to the minimal amount of complete genomes available and the lack of cytogenetic studies. *Acanthamoeba castellanii* (97) and *N. gruberi* (98) are most likely aneuploids, complicating the interpretation of population genetic analyses of nuclear loci. In addition, the impact of recombination on the evolutionary genetics of species of FLA is largely unknown and further obscured by the fact that, in most cases, the mode of inheritance and the impact of sexual reproduction are not well established (10, 99, 100).

Given these unknowns, and the advantages mentioned above, mitochondrial loci are valuable molecular markers to use for population genetic analyses of FLA.

In conclusion, this study significantly improves our knowledge of the diversity, distribution, and habitats colonized by *Naegleria* and *Vermamoeba*, providing valuable 18S rDNA, ITS, and COI reference sequences and optimized primers to generate them. To the best of our knowledge, this is the first work that used state-of-the-art Bayesian phylogenetics and single-locus species-delimitation tools combined with population genetics approaches to delimit species of FLA. However, the modest number of isolates obtained ($n = 32$) is a limiting aspect of this study. Although most of them (85.3%) grouped in either the *N. americana-N. galeacystis* or *V. vermiformis* complexes, the unexpectedly high number of cryptic species found within them reduces the number of isolates per ESU to just 2 to 10, which lies on the lower, suboptimal bound for species delimitation (46, 90). Finally, sequencing additional unlinked loci from the complete set of strains is required to harness the power of modern multilocus sequence analysis approaches based on the multispecies coalescent process (91, 101, 102).

## MATERIALS AND METHODS

**Sampling sites, sample collection, and isolation of free-living amoebae.** Eight sampling sites were selected (Table 1) within the three major ecosystems (by extension) found in the State of Morelos (103), Central Mexico, spanning an altitudinal and bioclimatic gradient (103). They comprise (i) the seasonally dry tropical forest, which covers the most significant area (about 52%) in the lower and warmer parts of the State, dominated by a high diversity of deciduous leguminous shrubs and trees (104); (ii) mixed pine-oak forests, which grow on the northern slopes of the State of Morelos; and (iii) pine-fir forests, which grow on the highest mountains of the state. The sites are characterized by umbric and mollic andosols, derived from volcanic materials, and temperate to mildly cold climates. In addition, three sites were from the municipal water supply system in Cuernavaca, the state's capital.

Samples of river water columns were collected in sterile 50-mL Falcon tubes. Superficial (3 to 5 cm) river sediment cores were obtained by digging 15-mL Falcon tubes, which had their lower end cut, into the riverbed. The soil samples were collected with a garden trowel and stored in Ziplock bags. Domestic shower heads and a water purifying system were sampled by scraping off biofilms with sterile cotton swabs, which were stored in 15-mL Falcon tubes with 2 mL Page's amoebal saline. Samples were processed on the collection day. FLA were isolated on nonnutrient agar (NNA) seeded with inactivated *Escherichia coli* (NAA-Ec) cultures (105). The plates were incubated at 30°C or 37°C and inspected every 2 to 3 days. Samples from the migration fronts were reinoculated onto new NAA-Ec plates for at least seven rounds until morphologically homogeneous cultures were obtained. Table 2 lists the 32 isolates.

**DNA extraction, PCR primers, and sequencing of amplicons.** Genomic DNA was extracted from 70 to 80% confluent cultures grown in vented T25 flasks with NNA-Ec and incubated at 25°C using a DNeasy ultra clean microbial kit (Qiagen), following the manufacturer's instructions. Three genetic markers were amplified using the primers listed in Table 3, as follows a fragment of the 18S rRNA gene spanning the V4 to V5 regions (22), the entire ITS1-5.8-ITS2 rDNA region (ITS), and internal fragments of the mitochondrial cytochrome c oxidase subunit I (COI) gene. In addition, the JITS primers designed to amplify the ITS region from Tetramitia (23) were adapted to target different groups of FLA, as indicated in Table 3. Likewise, the classic animal COI barcoding primers (24) were adapted to target several taxa of FLA. All the PCR products were purified with the GeneJET PCR purification kit, following the manufacturer's instructions, and sent to Macrogen Korea for Sanger sequencing of both strands. Contigs were assembled as described previously (87).

**Reference sequences for phylogenetic analysis.** Relevant reference sequences were retrieved from the NCBI nucleotide and taxonomy databases using the Entrez querying system (106) and BLASTN searches performed at the NCBI portal (107). The product of HaveOM_p35 (*H. vermiformis* Cox1/2) was used in a tblastn search against a locally formatted database of the unannotated mitochondrial DNA sequence from *V. vermiformis* isolate TW EDP1 (CM036233.1) to identify the coordinates of its *cox1* coding DNA sequence (CDS). The corresponding DNA segment was extracted and aligned to the complete *cox1* genes from the *Acanthamoeba castellanii* Neff (AccaoMp04), *Acanthamoeba polyphaga* Linc_Ap-1 (KP054475), *H. vermiformis* (HaveOM_p35), and *V. vermiformis* (AB846_56) complete mitochondrial genome sequences. The resulting alignment revealed that the TW EDP1 cox1 sequence contained multiple single T insertions in homopolymeric T runs, which are introduced by the Oxford Nanopore Technologies (ONT) sequencing technology used by the authors of the ASM2069642v1 assembly. After removal of these insertions, the intact CDS could be restored and was used for phylogenetic analysis.

**Phylogenetic analysis of SSU rDNA, ITS, and COI sequences.** The SSU rDNA sequences were aligned with SINA v1.7.2 (108), which aligns rRNA sequences to a reference multiple sequence alignment from SILVA release 138.1 (109). We used SeaView v5.0.5 (110) to remove gap-only columns and to trim the alignment borders. The ITS regions (ITS1-5.8S-ITS2) and partial COI sequences were aligned using MAFFT v7.453 (111) with the high accuracy mode (linsi).

Phylogenetic analyses were performed under maximum likelihood and Bayesian optimality criteria on curated multiple sequence alignments with identical sequences collapsed to haplotypes. We used IQTree2 v2.2.0 (112) for maximum likelihood tree searching, employing the Bayesian information criterion (BIC) for model selection with the aid of ModelFinder (113). Branch support values were computed

using the ultrafast bootstrap method implemented in IQTree2. We used BEAST 2.6.7 (114) to infer Bayesian phylogenies of ITS and COI sequences from *Naegleria* and *Vermamoeba* using the best model selected by ModelFinder (113) under BIC in IQTree2. Competing clock models were evaluated in a Bayesian framework using nested sampling (115) to compute marginal likelihoods (MLs) with the aid of the NS package v1.2 for BEAST 2, with 20 particles and a subchain length of 1e4. Bayes factors were computed from the MLs of competing models and interpreted according to Kass and Raftery (116). Each BEAST MCMC chain was run for $5 \times 10^7$ generations, sampling the posterior every 5,000th, with 3 replicate runs, summarizing results, determining effective sample sizes (ESSs) of parameter estimates, and determining the convergence of runs as described previously (91). The final 33% trees of each replicate MCMC chain were pooled with LogCombiner v2.6.7 and displayed as the maximum clade credibility (MCC) tree produced by TreeAnnotator v2.6.7 from the BEAST 2.6.7 suite.

**Single-locus species delimitation using distance-based (ASAP) and phylogeny-aware (mPTP) methods.** We took advantage of the recent development of improved distance-based (ASAP) and phylogeny-aware methods (mPTP) to delimit species based on a single locus to compare their performances in automatically delimiting species in the genera *Naegleria* and *Vermamoeba*. Assemble species by automatic partitioning (ASAP) implements a hierarchical clustering algorithm that uses only pairwise genetic distances, proposing species partitions ranked by a scoring system that uses no prior biological insight of intraspecific diversity (45). Each newly created partition is characterized in two complementary ways. First, it gets a probability assigned that quantifies the chances that each of its new groups is a single species. Second, the width of the barcode gap between the previous and the new partition is computed. Both metrics (probability and barcode gap width) are combined into a single ASAP score used to rank the partitions (45). ASAP for Linux was compiled from the source and provided with best-fit maximum likelihood distance matrices computed by IQTree2 or directly from alignments using the K2P model with default parameters. The mPTP algorithm (47) is a phylogeny-aware method based on the Poisson tree processes (PTP) algorithm, which tries to determine the transition point from a between- to a within-species process using a two-parameter model, with one for the speciation and one for the coalescent processes (117). The mPTP implementation extends the PTP model by using a distinct exponential distribution to fit the branching events within each delimited species. This parameter relaxes the constraint of assuming a single coalescence rate across the tree by accommodating intraspecies-specific coalescence rates (47). mPTP analyses were performed with v0.2.4 compiled from the source on a 64-bit Linux machine (https://github.com/Pas-Kapli/mptp). We completed 10 different runs for each data set with the following settings: performing an MCMC run for $100 \times 10^6$ generations, sampling the chain every 5,000 generations, and discarding the first 20,000 samples as burn-in. Convergence of the independent runs was assessed through the average standard deviation of delimitation support values (ASDDSV). The overall support for the ML estimate of species-like units was calculated as the mean of the average support values (ASVs) over the 10 runs. We mapped the ASAP and mPTP delimitations on the corresponding Bayesian trees with the aid of iTOL (118).

**Population genetic analysis of DNA sequence polymorphisms.** Within- and between-species DNA sequence polymorphisms were analyzed with DnaSP6.0 (119) to obtain estimates of population differentiation (48) and gene flow across them (96), as described previously (87). The analyses are based on the segregating sites, excluding those that violate the infinite site's model (i.e., those segregating two or more bases). In addition, permutation analyses were run (1e + 4 replicates) to test the significance of the population subdivision test statistics (48).

**Morphological characterization.** The morphological characterization of the strains was limited to assessing their morphotype, as recommended by Smirnov and Brown (64), employing an inverted microscope (AxioVert.A1; Zeiss) equipped with long-distance phase contrast and differential interference contrast for plastic (PlasDIC) objectives ($40\times$ and $63\times$) and filters. Images of living cultures in Page's Amoeba Saline (PAS) supplemented with heat-killed *E. coli* DH5$\alpha$ incubated in a vented T25 cell-culture flask at 25°C were captured with an Axiocam 202 mono camera (Zeiss).

**Statistical (association) analysis.** To test the null hypothesis of nonassociation between genus or species with habitat and ecosystem, a formal three-way association test of categorical variables (120) was performed using the R packages vcd (121, 122) and vcdExtra (123).

**Data availability.** All sequences generated in this study have been deposited in GenBank under accession numbers OP297004 to OP297035, OP454446 to OP454461, OP466812 to OP466823, and OP501834 to OP501855. Isolates are maintained in the corresponding author's laboratory.

## ACKNOWLEDGMENTS

We thank Alfredo Hernández and Víctor del Moral for their expert technical support with server maintenance at the Centro de Ciencias Genómicas, UNAM. We also thank Fabiola Miranda-Sánchez for her critical reading of the manuscript.

We gratefully acknowledge the financial support received from the Consejo Nacional de Ciencia y Tecnología (CONACyT Mexico, A1-S-11242) and Programa de Apoyo a Proyectos de Investigación e Innovación Tecnológica (PAPIIT) from the Universidad Nacional Autónoma de México (DGAPA-PAPIIT, UNAM, IN209321).

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
