## [Reviewer comments · Microbiology Spectrum]

Microbiology Spectrum

Extensive cryptic diversity and ecological associations uncovered among Mexican and global collections of *Naegleria* and *Vermamoeba* by 18S rDNA, ITS, and COI sequence analysis

Juan Zurita-Artaloitia, Javier Rivera, and Pablo Vinuesa

Corresponding Author(s): Pablo Vinuesa, Universidad Nacional Autonoma de Mexico - Campus Morelos

Review Timeline:

Submission Date:	September 23, 2022
Editorial Decision:	November 25, 2022
Revision Received:	February 4, 2023
Accepted:	February 26, 2023

Editor: Adriana Lopes dos Santos

Reviewer(s): The reviewers have opted to remain anonymous.

Transaction Report:

DOI: <https://doi.org/10.1128/spectrum.03795-22>

November 25, 2022

Prof. Pablo Vinuesa
Centro de Ciencias Genómicas, UNAM
Programa de Ingeniería Genómica
Av. Universidad s/n
Col. Chamilpa
Cuernavaca, Mor. 62210
Mexico

Re: Spectrum03795-22 (**Extensive cryptic diversity and ecological associations uncovered among Mexican and global collections of *Naegleria* and *Vermamoeba* by 18S rDNA, ITS, and COI sequence analysis**)

Dear Prof. Pablo Vinuesa:

Thank you for submitting your manuscript to Microbiology Spectrum.

The overall text needs a revision. Please avoid long and extensive paragraphs and break the sentences for clarity. Also check for none standard terms (e.g. new mexican) or unclear codes (e.g. PopSet 399764... line 123) throughout the text.

Also, please include more information both in the introduction and discussion about the ecology of the studied genera. Particularly, your introduction and abstract are focused in listing what you did rather than the ecology of these organisms.

Any supplemental material should be submitted with the paper as separate section. This section of the paper should include legends for any supplemental material that is intended for posting. Please only use unstructured repository such as figshare for large files, for example supplementary table 1. Also add to this repository all alignments and/or text file with the trees generate. All the links to figshare cited throughout the text should be on data availability. The link available on this section is not functional.

A statement about the availability of your isolates should also be added. Have they been deposited in a local culture collection?

Link Not Available

Sincerely,

Adriana Lopes dos Santos

Journals Department
Reviewer comments:

Staff Comments:

Preparing Revision Guidelines

Please return the manuscript within 60 days; if you cannot complete the modification within this time period, please contact me. If you do not wish to modify the manuscript and prefer to submit it to another journal, please notify me of your decision immediately so that the manuscript may be formally withdrawn from consideration by Microbiology Spectrum.

62. You could clarify that you're referring to FLA species/genus (which many may be considered broadly distributed erroneously due to the issues you mention), not FLA as whole group (which is a wide group, broadly distributed).

68. Recent works by Gonzalez-Miguens et al. used COI in Arcellinida testate amoebas ("González-Miguéns, Rubén, et al. "Multiple convergences in the evolutionary history of the testate amoeba family Arcellidae (Amoebozoa: Arcellinida: Sphaerothecina): when the ecology rules the morphology." *Zoological Journal of the Linnean Society* 194.4 (2022): 1044-1071.", and "González-Miguéns, Rubén, et al. "Deconstructing Diffugia: The tangled evolution of lobose testate amoebae shells (Amoebozoa: Arcellinida) illustrates the importance of convergent evolution in protist phylogeny." *Molecular Phylogenetics and Evolution* 175 (2022): 107557."), he also generated a metabarcoding protocol for them based on that gene (Gonzalez-Miguens, Ruben, et al. "A needle in a haystack: a new metabarcoding approach to survey diversity at the species level of Arcellinida (Amoebozoa: Tubulinea)." *bioRxiv* (2022)), maybe you can include the reference. Although you don't need to modify the sentence, I agree that considering protists and FLA as a whole, the use of COI is marginal.

92. The new Mexican what? It took me a couple reads to understand you meant your new Mexican sequences. Maybe it's more clear something like this: "...population genetic analysis **with** selected global reference sequences confirmed the identification of **our** isolates at the species level..." or "...analysis of our new Mexican sequences with others selected from global reference databases...".

Intro, general comments: It's correct, but maybe you could talk a bit more about these organism and their ecological importance

170. You could mention what implications does it have that the null of independence is rejected. Or talk a bit more about the ecology in the discussion.

551. Just a small grammar error, I think: "...identify and exclude...". If that's not that what you meant, the sentence may need restructuring, because it's confusing.

Point-by-point responses to the reviewer's inquiries:

The authors thank the reviewer for the constructive criticism, which helped us improve the manuscript. Line numbering corresponds to the Marked-Up_Manuscript.pdf file.

62. This issue was clarified. The sentence was re-structured as follows, providing selected references to sustain the statement:

L88-91.

FLA species are generally thought to be broadly distributed (16, 17). However, poorly resolving molecular markers, combined with a shallow sampling of a few ecosystems, constrain our knowledge of the diversity and ecological features of FLA on local, regional, and global geographical scales (18–20).

68. These are all excellent and relevant references added to the manuscript. However, they are cited in the discussion to avoid excessive citations in the Introduction. For example, these references (73–75) nicely fit the last sentences of the first paragraph of the discussion.

L529-534

However, it is well established that morphological characters, even for testate amoebae like the well-studied Arcellinida, are often misleading for reliable taxonomic identification of FLA due to their homoplasious nature (73, 74). Furthermore, amoebal morphospecies typically hide multiple cryptic species (14, 18, 75).

The reference to the COI-metabarcoding paper was cited again at the end of the discussion section on COI markers.

L694-700

This analysis revealed that the analyzed COI segment is significantly more polymorphic than the ITS region and, therefore, better suited for ecological inference and molecular systematics studies. Furthermore, COI markers pave the way for finer-grained association studies between FLA lineages and environmental variables, particularly if coupled with high-throughput sequencing of lineage-specific COI-barcodes generated directly from environmental DNA, as recently reported (75).

92. This problematic sentence was rephrased for clarity as follows:

L119-121

Combined phylogenetic and population genetic analyses of the new sequences generated in this study and selected global reference sequences confirmed the identification of the isolates at the species level based on the current taxonomy.

170. The following sentence was added to explicitly provide our biological interpretation of the statistical test:

L200-202

This result indicates that the sampled organisms from both genera have strongly differentiated environmental distributions, which may result from distinct habitat preferences or adaptations.

551. The problematic sentence (last one in the following paragraph, provided for context) was corrected (identifying instead of to identify) and improved (shortened and avoiding passive voice) as follows:

L590-598

However, we found that both delimitation methods require some data pre-processing to perform optimally, like collapsing sequences to haplotypes. In addition, mPTP requires at least two haplotypes per ESU, as the shifting point between speciation and coalescent processes cannot be determined for singleton species or units because no coalescence events are available for them in the dataset (46). Therefore, we paid great attention to identifying and excluding singleton units from datasets to be analyzed with mPTP to avoid lumping these lineages with sister units or clades.

February 26, 2023

Prof. Pablo Vinuesa
Universidad Nacional Autonoma de Mexico - Campus Morelos
Centro de Ciencias Genómicas
Av. Universidad s/n
Col. Chamilpa
Cuernavaca, Mor. 62210
Mexico

Re: Spectrum03795-22R1 (**Extensive cryptic diversity and ecological associations uncovered among Mexican and global collections of *Naegleria* and *Vermamoeba* by 18S rDNA, ITS, and COI sequence analysis**)

Dear Prof. Pablo Vinuesa:

Your manuscript has been accepted, and I am forwarding it to the ASM Journals Department for publication. You will be notified when your proofs are ready to be viewed.

Sincerely,

Adriana Lopes dos Santos
Editor, Microbiology Spectrum
